# A Kv2 inhibitor combination reveals native neuronal conductances consistent with Kv2/KvS heteromers

Robert G Stewart[1,2,3], Matthew James Marquis[1], Sooyeon Jo[3], Brandon J Harris[1], Aman S Aberra[2,4], Verity Cook[2,5], Zachary Whiddon[2], Vladimir Yarov-Yarovoy[1,6], Michael Ferns[1,6], Jon T Sack[1,2,6]*

[1]Department of Physiology and Membrane Biology, University of California Davis, Davis, United States; [2]Neurobiology Course, Marine Biological Laboratory, Woods Hole, United States; [3]Department of Neurobiology, Harvard Medical School, Boston, United States; [4]Department of Biological Sciences, Dartmouth College, Hanover, United States; [5]Einstein Center for Neuroscience, Charité Universitätsmedizin Berlin, Hufelandweg, Germany; [6]Department of Anesthesiology and Pain Medicine, University of California Davis, Davis, United States

*For correspondence: jsack@ucdavis.edu

## eLife Assessment

Some delayed rectifier currents in neurons are formed by the combination of Kv2 and silent subunits, KvS. However, we lack the tools to identify these heteromeric channels in vivo. In this **important** study by the Sack group, the authors identify a pharmacological tool that can reveal the presence of KvS subunits as components of the delayed rectifier potassium currents in selected neurons. The experimental evidence presented in the manuscript is **compelling** and represents a significant advance that should be of interest to a wide community of neuroscientists and channel physiologists.

**Abstract** KvS proteins are voltage-gated potassium channel subunits that form functional channels when assembled into heteromers with Kv2.1 (*KCNB1*) or Kv2.2 (*KCNB2*). Mammals have 10 KvS subunits: Kv5.1 (*KCNF1*), Kv6.1 (*KCNG1*), Kv6.2 (*KCNG2*), Kv6.3 (*KCNG3*), Kv6.4 (*KCNG4*), Kv8.1 (*KCNV1*), Kv8.2 (*KCNV2*), Kv9.1 (*KCNS1*), Kv9.2 (*KCNS2*), and Kv9.3 (*KCNS3*). Electrically excitable cells broadly express channels containing Kv2 subunits and most neurons have substantial Kv2 conductance. However, whether KvS subunits contribute to these conductances has not been clear, leaving the physiological roles of KvS subunits poorly understood. Here, we identify that two potent Kv2 inhibitors, used in combination, can distinguish conductances of Kv2/KvS heteromers and Kv2-only channels. We find that Kv5, Kv6, Kv8, or Kv9-containing channels are resistant to the Kv2-selective pore-blocker RY785 yet remain sensitive to the Kv2-selective voltage sensor modulator guangxitoxin-1E (GxTX). Using these inhibitors in mouse superior cervical ganglion neurons, we find predominantly RY785-sensitive conductances consistent with channels composed entirely of Kv2 subunits. In contrast, RY785-resistant but GxTX-sensitive conductances consistent with Kv2/KvS heteromeric channels predominate in mouse and human dorsal root ganglion neurons. These results establish an approach to pharmacologically distinguish conductances of Kv2/KvS heteromers from Kv2-only channels, enabling investigation of the physiological roles of endogenous KvS subunits. These findings suggest that drugs which distinguish KvS subunits could modulate electrical activity of subsets of Kv2-expressing cell types.

## Introduction

The Kv2 voltage-gated K$^+$ channel subunits, Kv2.1 and Kv2.2, are broadly expressed in electrically excitable cells throughout the body and have important ion-conducting and non-conducting functions (*Trimmer, 1993*; *Du et al., 2000*; *Li et al., 2013*; *Liu and Bean, 2014*; *Bishop et al., 2015*; *Johnson et al., 2018*; *Kirmiz et al., 2018*; *Vierra et al., 2021*; *Matsumoto et al., 2023*). Consistent with this widespread expression, Kv2 channels have profound impacts on many aspects of our physiology including vision, seizure suppression, stroke recovery, pain signaling, blood pressure, insulin secretion, and reproduction (*Bocksteins, 2016*). Although modulation of Kv2 channels may hold therapeutic promise, Kv2 subunits are poor systemic drug targets due to their importance in many tissues.

A potential source of molecular diversity for Kv2 channels are a group of Kv2-related proteins which have been referred to as regulatory, silent, or KvS subunits (*Bocksteins et al., 2009*; *Kobertz, 2018*). KvS subunits are an understudied class of voltage-gated K$^+$ channel (Kv) subunits that comprise one fourth of mammalian Kv subunit types. Like all other Kv proteins, the ten KvS proteins (Kv5.1, Kv6.1–6.4, Kv8.1–8.2, and Kv9.1–9.3) are alpha subunits with a voltage sensor and pore domain. Distinct from other Kv alpha subunits, KvS have not been found to form functional homomeric channels. Rather, KvS alpha subunits co-assemble with Kv2 alpha subunits to form heterotetrameric Kv2/KvS channels in which the KvS subunit makes up part of the K$^+$-conductive pathway (*Salinas et al., 1997b*; *Kramer et al., 1998*). Kv2/KvS heteromers have biophysical properties distinct from those of homomeric Kv2 channels (*Post et al., 1996*; *Salinas et al., 1997a*; *Kramer et al., 1998*; *Richardson and Kaczmarek, 2000*; *Zhong et al., 2010*; *Bocksteins et al., 2012*; *Bocksteins et al., 2017*). KvS mRNAs are expressed in tissue and cell-specific manners that overlap with Kv2.1 or Kv2.2 expression (*Castellano et al., 1997*; *Salinas et al., 1997a*; *Kramer et al., 1998*; *Bocksteins et al., 2012*; *Bocksteins and Snyders, 2012*; *Bocksteins, 2016*). These expression patterns and functional effects suggest that Kv2 conductances in many cell types might be Kv2/KvS heteromeric conductances. Consistent with narrow expression of the many KvS subunits, genetic mutations and gene-targeting studies have linked disruptions in the function of different KvS subunits to defects in distinct organ systems including retinal cone dystrophy (*Wu et al., 2006*; *Hart et al., 2019*; *Inamdar et al., 2022*), male infertility (*Regnier et al., 2017*), seizures (*Jorge et al., 2011*), and changes in pain sensitivity (*Tsantoulas et al., 2018*). These organ-specific disruptions suggest that each KvS subunit selectively modulates a subset of Kv2 channels. However, studies of the physiological roles of KvS subunits have been hindered by a lack of tools to identify native KvS conductances. Due to limited KvS pharmacology, there is little evidence that definitively ascribes native K$^+$ conductances to KvS-containing channels. While studies have identified native conductances attributed to KvS subunits (reviewed by *Bocksteins, 2016*), it has not been clear whether the Kv2 conductances that are prominent in many electrically-excitable cell types are carried by Kv2-only channels, or Kv2/KvS heteromeric channels.

No drugs are known to be selective for KvS subunits. However, Kv2/KvS heteromeric channels do have some pharmacology distinct from channels that contain only Kv2 subunits. Quaternary ammonium compounds, 4-aminopyridine, and other broad-spectrum K$^+$ channel blockers have different potencies against certain KvS-containing channels as compared to Kv2 channels (*Post et al., 1996*; *Thorneloe and Nelson, 2003*; *Stas et al., 2015*). However, these blockers are poorly selective and cannot effectively isolate Kv2/KvS conductances from the many other voltage-gated K$^+$ conductances of electrically excitable cells.

Highly selective Kv2 channel inhibitors fall into two mechanistically distinct classes. One class is the inhibitory cystine knot peptides from spiders. An exemplar of this class is the tarantula toxin guangxi-toxin-1E (GxTX), which has remarkable specificity for Kv2 channel subunits over other voltage-gated channels (*Herrington et al., 2006*; *Thapa et al., 2021*). GxTX binds to the voltage sensor of each Kv2 subunit (*Milescu et al., 2009*), and stabilizes that voltage sensor in a resting state to prevent channel opening (*Tilley et al., 2019*). GxTX binding requires a specific sequence of residues, TIFLTES, at the extracellular end of the Kv2 subunit S3 transmembrane helix (*Milescu et al., 2013*). This GxTX-binding sequence is conserved between Kv2 channels but is not retained by any KvS subunit. A second class of selective Kv2 inhibitors is a family of small molecules discovered in a high throughput screen for use-dependent Kv2 inhibitors (*Herrington et al., 2011*). Of these, RY785 is the most selective for Kv2 channels over other channel types. RY785 acts like a pore blocker which binds in the central cavity of Kv2 channels (*Marquis and Sack, 2022*). The central cavity-lining residues of all KvS subunits have

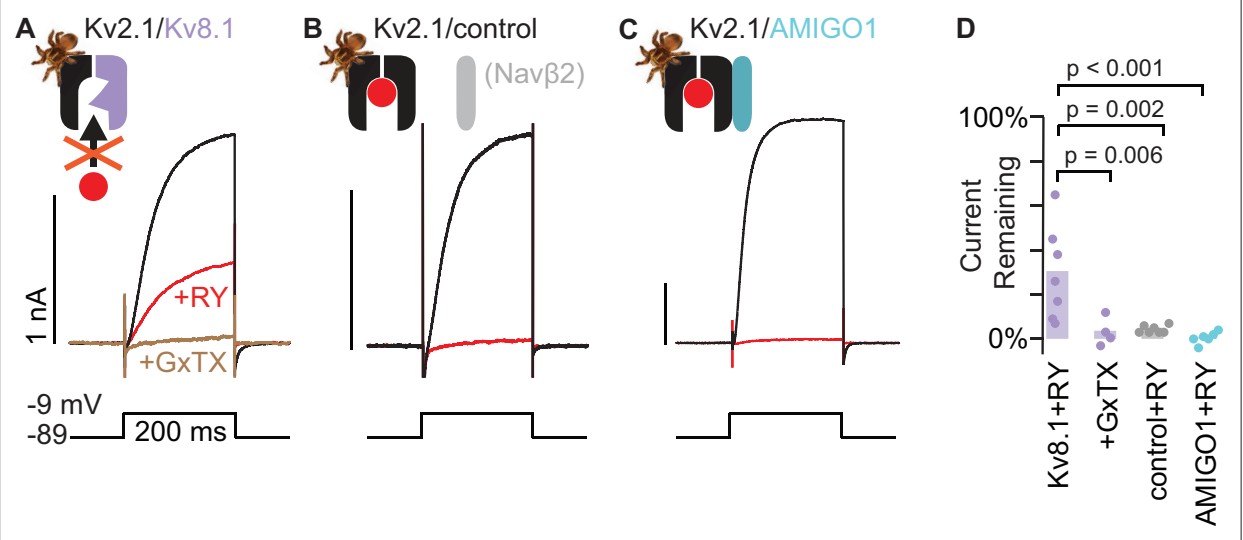

**Figure 1.** Kv2.1/Kv8.1 heteromers are resistant to RY785 and sensitive to GxTX. (**A**) Exemplar traces from a voltage-clamped Kv2.1-CHO cell transfected with Kv8.1. Black and red traces are currents before and after application of 1 µM RY785 respectively. Brown trace is current after subsequent application of 1 µM RY785 and 100 nM GxTX. (**B**) Exemplar traces from a Kv2.1-CHO cell transfected with Navβ2. (**C**) Exemplar traces from a Kv2.1-CHO cell transfected with AMIGO1. (**D**) Current remaining after application of 1 µM RY785 or 1 µM RY785 +100 nM GxTX. Bars represent mean. Each point represents current from one cell at the end of a 200ms voltage step to –9 mV. Dunnett tests with Kv8.1+RY785 as control.

differences from Kv2 subunits. We recently reported that coexpression of Kv5.1 with Kv2.1 led to a conductance that was resistant to RY785 (*Ferns et al., 2025*).

In this study, we develop a method to isolate conductances of KvS-containing channels. We identify that the combination of GxTX and RY785 can distinguish conductances of Kv2-only channels from channels that contain the KvS subtypes, Kv5, Kv6, Kv8, or Kv9. To determine whether cell types enriched with KvS mRNA have functional KvS-containing channels, we use these inhibitors to reveal native neuronal conductances consistent with Kv2/KvS heteromers in mouse and human dorsal root ganglion neurons. While this study does not address the impact of GxTX or RY785 on action potentials or in vivo, the distinct pharmacology of Kv2/KvS heteromers presented here suggests that KvS conductances could be targeted to selectively modulate discrete subsets of cell types.

## Results

### Kv2.1/Kv8.1 heteromers are resistant to RY785 and sensitive to GxTX

To identify a pharmacological strategy to distinguish Kv2/KvS heteromeric conductances from other endogenous neuronal conductances, we determined the Kv2/KvS selectivity of known Kv2 inhibitors. To test whether Kv2 inhibitors also inhibit Kv2/KvS heteromeric channels, we transfected KvS cDNA into a stable cell line which was subsequently induced to express Kv2.1 (Kv2.1-CHO) and later recorded whole cell currents. We previously found that 1 µM RY785 or 100 nM GxTX blocked almost all the voltage-gated K$^+$ conductance of this Kv2.1-CHO cell line, with 1±2% or 0 ± 0.1% (mean ± SEM) current remaining respectively at 0 mV (*Tilley et al., 2019*; *Marquis and Sack, 2022*). To test the pharmacological response of KvS we began with Kv8.1, a subunit that creates heteromers with biophysical properties distinct from Kv2 homomers (*Salinas et al., 1997a*), and modulates motor neuron vulnerability to cell death (*Huang et al., 2024*). After transfection of Kv8.1 into Kv2.1-CHO cells, we find that a sizable component of the delayed rectifier current became resistant to 1 µM RY785 (*Figure 1A*). This RY785-resistant current was inhibited by 100 nM GxTX, suggesting that the GxTX-sensitivity arises from inclusion of Kv2.1 subunits in the channels underlying the RY785-resistant current. A simple interpretation is that RY785-resistant yet GxTX-sensitive currents are carried by Kv2.1/Kv8.1 heteromeric channels. The fraction of RY785-resistant current had a pronounced cell-to-cell variability (*Figure 1D*). Co-expression of KvS and Kv2 subunits can result in Kv2 homomers and

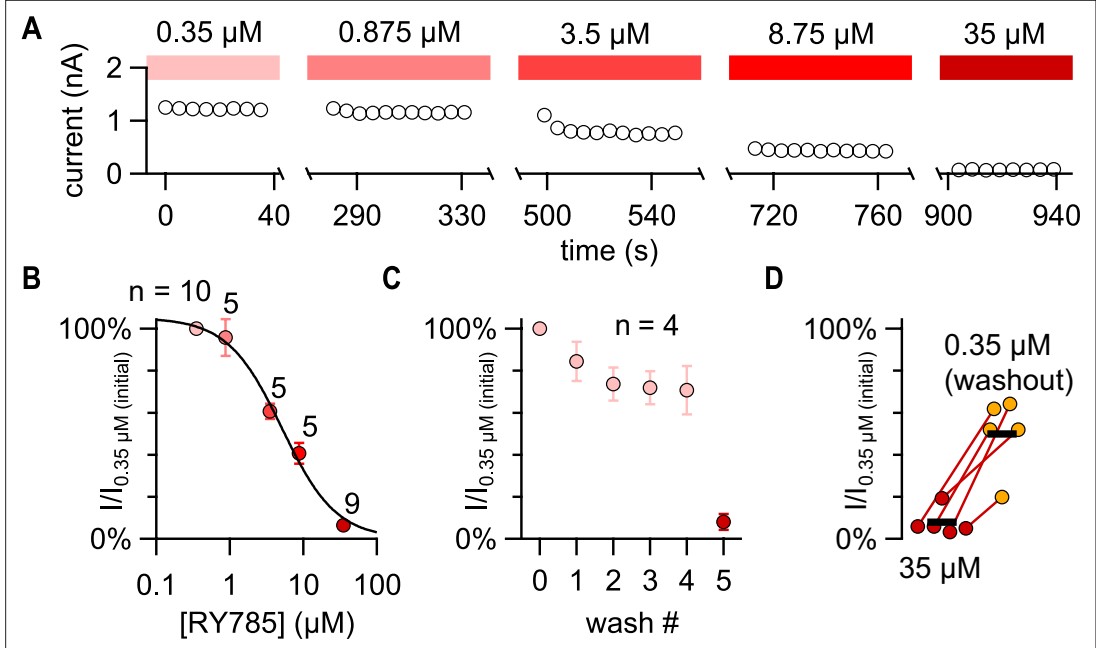

**Figure 2.** RY785 blocks Kv2.1/Kv8.1 heteromers in a concentration-dependent manner. (**A**) Current amplitudes during an RY785 concentration-effect experiment on a Kv2.1-CHO cell transfected with Kv8.1. Circles represent tail current amplitudes 2–4ms into a step to –9 mV following a 200ms activating step to 71 mV. Voltage protocol was repeated in 5 s intervals. Solution exchanges occurred during the gaps in the time axis. For exemplar current traces, see *Figure 3*, *Figure 2—figure supplement 1*. (**B**) Mean normalized tail current amplitudes with increasing concentrations of RY785. Error bars represent SEMs. Black curve is a fitted Hill function with $n_H$=1 (IC$_{50}$=5.1 ± 1.0 µM, base = 1.0 ± 0.1 %). (**C**) Vehicle control tail current with repeated solution exchanges (washes) into 0.35 µM RY785, mimicking solution exchanges in panel B. Vehicle control solution exchanges were followed by exchange into 35 µM RY785 (wash #5). Error bars represent SEMs from n=4 cells. (**D**) Tail current recovery following solution exchange from 35 µM RY785 into 0.35 µM RY785 (washout). Bars represent mean current amplitudes from n=5 cells.

The online version of this article includes the following figure supplement(s) for figure 2:

**Figure supplement 1.** Rapid unblock of RY785 from Kv2.1/Kv8.1 heteromers.

Kv2/KvS heteromers (*Pisupati et al., 2018*), and we presume variability in the RY785-sensitive fraction results from cell-to-cell variability in the proportion of Kv8.1-containing channels.

As a control, we transfected Kv2.1-CHO cells with Navβ2, a transmembrane protein not expected to interact with Kv2.1. In Kv2.1-CHO cells transfected with Navβ2, 1 µM RY785 efficiently blocked Kv2.1 conductance, leaving 4 ± 0.6% (mean ± SEM) of current (*Figure 1B and D*). We also transfected Kv2.1-CHO cells with a member of the AMIGO family of Kv2-regulating transmembrane proteins. AMIGO1 promotes voltage sensor activation of Kv2.1 channels in these Kv2.1-CHO cells (*Sepela et al., 2022*). 1 µM RY785 blocked Kv2.1 conductances in cells transfected with AMIGO1, leaving 0.6 ± 1% (SEM) of current (*Figure 1C and D*). These control experiments indicate that transfection of a set of other transmembrane proteins did not confer resistance to RY785, suggesting that the RY785 resistance is not generically induced by overexpression of non-KvS transmembrane proteins.

To determine whether Kv8.1-containing channels are completely resistant to RY785, we performed an RY785 concentration-effect experiment. To pre-block Kv2.1 homomers, we began concentration-effect measurements at 0.35 µM RY785, which we expect to block 98% of homomers based on the estimated $K_D$ of 6 nM RY785 for the Kv2.1 currents in these Kv2.1-CHO cells (*Marquis and Sack, 2022*). Notably, currents resistant to 0.35 µM RY785 were blocked by higher concentrations of RY785, with nearly complete block observed in 35 µM RY785 (*Figure 2A*). We quantified block of tail currents at –9 mV following a 200ms step to +71 mV, and normalized to current from the initial 0.35 µM RY785 treatment. This protocol revealed an IC$_{50}$ of 5±1 µM (SD; *Figure 2B*). The Hill coefficient of 1.2±0.2 is consistent with 1:1 binding to a homogenous population of RY785-inhibited channels. We had noted that solution exchanges can change current amplitudes, and interleaved time-matched solution exchange controls. These controls revealed variable current rundown of approximately 30% on average (*Figure 2C*). Time-matched control washes were followed by treatment with 35 µM RY785

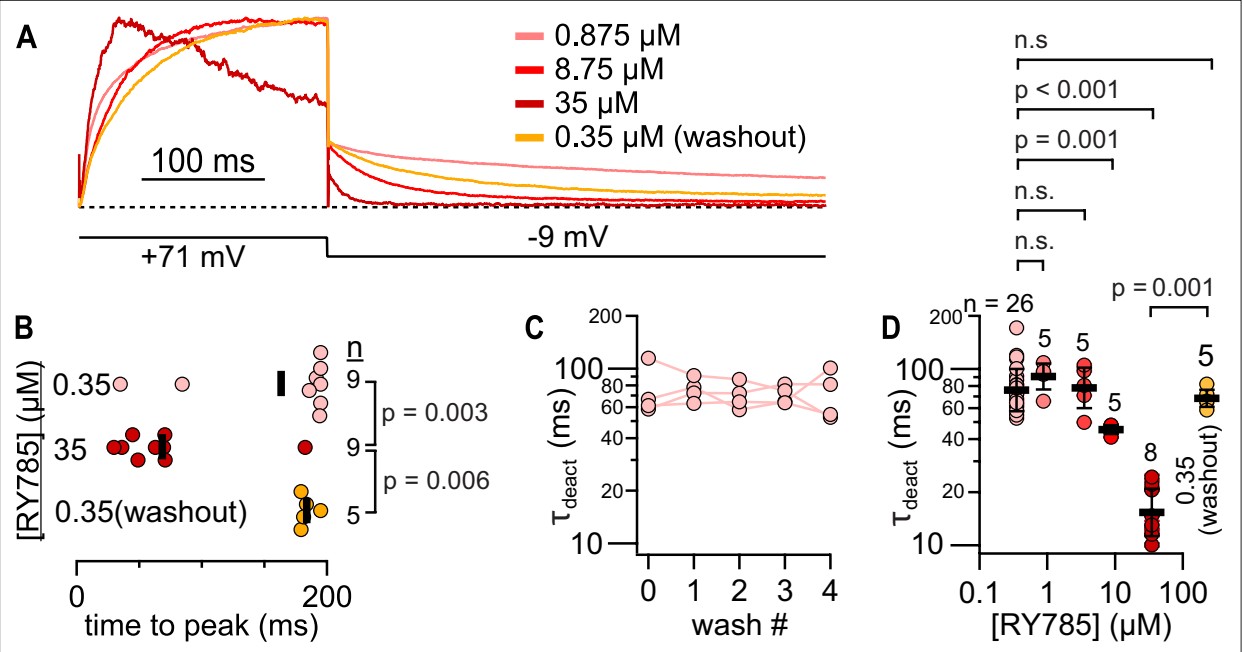

**Figure 3.** RY785 can affect Kv2.1/Kv8.1 current kinetics. (**A**) Kinetics of currents from a Kv2.1-CHO cell transfected with Kv8.1 are altered by RY785. Traces normalized to max. (**B**) Latency to peak current during steps to +71 mV. The time axis of this plot is aligned with that of Panel A. Bars represent means. Unpaired Wilcoxon rank tests. (**C**) Time constant of deactivation at –9 mV is constant after washes with 0.35 µM RY785. Time constants are derived from fits of a monoexponential function (**Equation 3** with $A_2$ set equal to 0) to tail currents like those shown in Panel A. Fits were from the peak of each tail current to 200ms after the voltage step. Brown and Forsythe test p=0.98. ANOVA p=0.98. Statistics were performed on natural logarithms of time constants. n=4 cells. (**D**) RY785 can alter time constant of deactivation. Bars represent means. Brown and Forsythe test p=$3 \times 10^{-12}$. Unpaired Welch test p=$1 \times 10^{-7}$. Dunnett tests with 0.35 µM RY785 (initial) as control. Unpaired Wilcoxon rank test comparing 35 µM RY785 to 0.35 µM RY785 (washout) p=0.001. Statistics were performed on natural logarithms of time constants.

to confirm that currents in these cells had similar RY785 sensitivity to those in our concentration-effect experiment. Following block by 35 µM RY785, washing with 0.35 µM RY785 caused increases in current amplitudes in every trial (**Figure 2D**). The run-down and incomplete wash-out indicates that the 5 µM $IC_{50}$ of RY785 for these resistant channels may be an underestimate. Kv2.1/Kv8.1 currents were unblocked in the first current test following RY785 washout (**Figure 2—figure supplement 1**). This rapid recovery indicates that unblocking of Kv2.1/Kv8.1 heteromers occurred during the less than 3 min wash time at –89 mV, or that RY785 unblocked on the millisecond time scale during activating voltage pulses. This is distinct from Kv2.1 homomers, where RY785 becomes trapped in deactivated channels and unblocks much more slowly, with a time constant of about 2 hr at –92 mV or 100 s at +28 mV (**Marquis and Sack, 2022**). The dramatically faster unblock from Kv2.1/Kv8.1 is consistent with the weaker affinity observed for RY785. Overall, the results suggest that Kv8.1-containing channels in these Kv2.1-CHO cells form a pharmacologically homogenous population with an affinity for RY785 ~3 orders of magnitude weaker than Kv2.1 homomers. Our estimates of the affinities of Kv2.1 homomeric and Kv2.1/Kv8.1 heteromeric channels for RY785 suggest that ~1 µM RY785 elicits nearly complete block of Kv2.1 homomer conductance while blocking little Kv2.1/KvS heteromer conductance.

## Biophysical properties of RY785-resistant conductance are consistent with Kv2.1/Kv8.1 channels

We wondered whether RY785 block of Kv2 homomers could better reveal the gating of heteromers. Previous studies have identified that Kv2/Kv8.1 gating kinetics are distinct from Kv2 homomers (**Salinas et al., 1997a**), and we set out to identify whether RY785-resistant currents from Kv8.1 transfected CHO cells had similarly modulated kinetics. While concentrations of RY785 that partially block Kv2.1/Kv8.1 modified the kinetics of voltage-dependent gating, suggesting state-dependent block, we did not observe modification of kinetics with 3.5 µM or lower concentrations of RY785 (**Figure 3**).

To study the biophysical properties of the Kv8.1 conductance in the Kv2.1-CHO cells, we analyzed currents in 1 µM RY785 to block the Kv2.1 homomers. For comparison, Kv2.1-CHO cells were transfected with a control plasmid and treated with a DMSO vehicle control. We stepped cells to –9 mV from a holding potential of –89 mV and fit the current rise with an exponential function (*equation 1*). Cells transfected with Kv8.1 and blocked with 1 µM RY785 had a significantly slower activation time constant and lower sigmoidicity (shorter relative activation delay) than those only expressing Kv2.1 (*Figure 4A–C*). Conductance-voltage relations were fit with a Boltzmann function (*equation 2*) revealing that the half-maximal conductance of Kv8.1-transfected cells is shifted positive relative to Kv2.1 alone (*Figure 4D and E*). We did not detect a significant difference in the steepness (z) of the conductance-voltage relation (*Figure 4F*). Currents from Kv8.1-transfected cells inactivated less during a 10 s step to –9 mV (*Figure 4G and H*). However, the steady-state inactivation of Kv8.1-transfected cells was shifted to more negative voltages and is less steep than Kv2.1-transfected cells (*Figure 4I–K*). We did not observe substantial changes in current amplitude in cells treated with DMSO vehicle control (*Figure 4—figure supplement 1*). Overall, the biophysical properties reported here are consistent with a previous report which identified that co-expression of Kv8.1 with Kv2.1 in *Xenopus* oocytes slows the rate of activation, reduces inactivation and shifts steady-state inactivation to more negative voltages (*Salinas et al., 1997a*). This previous report also identified a positive shift in the conductance-voltage relation when Kv8.1 is co-expressed with a Kv2 subunit (Kv2.2), similar to our findings with Kv2.1/Kv8.1. Together these results show that RY785-resistant currents in cells transfected with Kv8.1 are distinct from Kv2.1 homomer currents and have changes in gating consistent with prior reports of Kv8.1/Kv2 biophysics. This validates using RY785 block of Kv2 homomers as a method to reveal the biophysics of a Kv8.1-containing population.

## A subunit from each KvS subtype is resistant to RY785 but sensitive to GxTX

To test if RY785 resistance and GxTX sensitivity of Kv8.1 is shared broadly by KvS subunits, we similarly assessed subunits of Kv5, Kv6, and Kv9 subtypes: Kv5.1, Kv6.4, and Kv9.3. Each of these KvS subunits create Kv2/KvS heteromers that have distinct biophysical properties (*Kramer et al., 1998*; *Kerschensteiner and Stocker, 1999*; *Bocksteins et al., 2012*). Kv5.1/Kv2.1 heteromers play an important role in controlling the excitability of mouse urinary bladder smooth muscle (*Malysz and Petkov, 2020*), mutations in Kv6.4 have been shown to influence human labor pain (*Lee et al., 2020*), and deficiency of Kv9.3 disrupts parvalbumin interneuron physiology in mouse prefrontal cortex (*Miyamae et al., 2021*). We transfected Kv5.1, Kv6.4, and Kv9.3, to determine whether they also produced delayed rectifier current resistant to 1 µM RY785 yet sensitive to 100 nM GxTX (*Figure 5A–C*). We observed that, in 1 µM RY785,>10% of the voltage-gated current remained in 12/13 Kv5.1, 9/14 Kv6.4, and 5/5 Kv9.3 transfected cells (*Figure 5D*). Like Kv8.1, the fraction of RY785-resistant current had pronounced cell-to-cell variability (*Figure 5D*) suggesting that the RY785-sensitive fraction could be due to different ratios of functional Kv2.1 homomers to Kv2.1/KvS heteromers. Addition of 100 nM GxTX blocked RY785-resistant current from cells transfected with each of these KvS subunits. A slightly higher fraction of Kv9.3 current remained in 100 nM GxTX, possibly due to Kv9.3 negatively shifting the midpoint of the conductance voltage relationship (*Kerschensteiner and Stocker, 1999*). While a fraction of KvS subunits appear to be retained intracellularly, immunofluorescence for Kv5.1, Kv9.3, and Kv2.1 also appeared localized to the perimeter of transfected Kv2.1-CHO cells (*Figure 5—figure supplement 1*). These results show that voltage-gated outward currents in cells transfected with members from each KvS subtype have decreased sensitivity to RY785 but remain sensitive to GxTX. While we did not test every KvS subunit, the ubiquitous resistance suggests that all KvS subunits may provide resistance to 1 µM RY785 yet remain sensitive to GxTX, and that RY785 resistance is a hallmark of KvS-containing channels.

## The Kv2 conductances of mouse superior cervical ganglion neurons do not have KvS-like pharmacology

We set out to assess whether RY785 together with GxTX could be a means of distinguishing endogenous Kv2/KvS heteromers from Kv2 channels in native neurons. We first designed experiments to test whether RY785 could inhibit endogenous Kv2 currents in mice, by studying neurons unlikely to express KvS subunits. Rat superior cervical ganglion (SCG) neurons have robust GxTX-sensitive

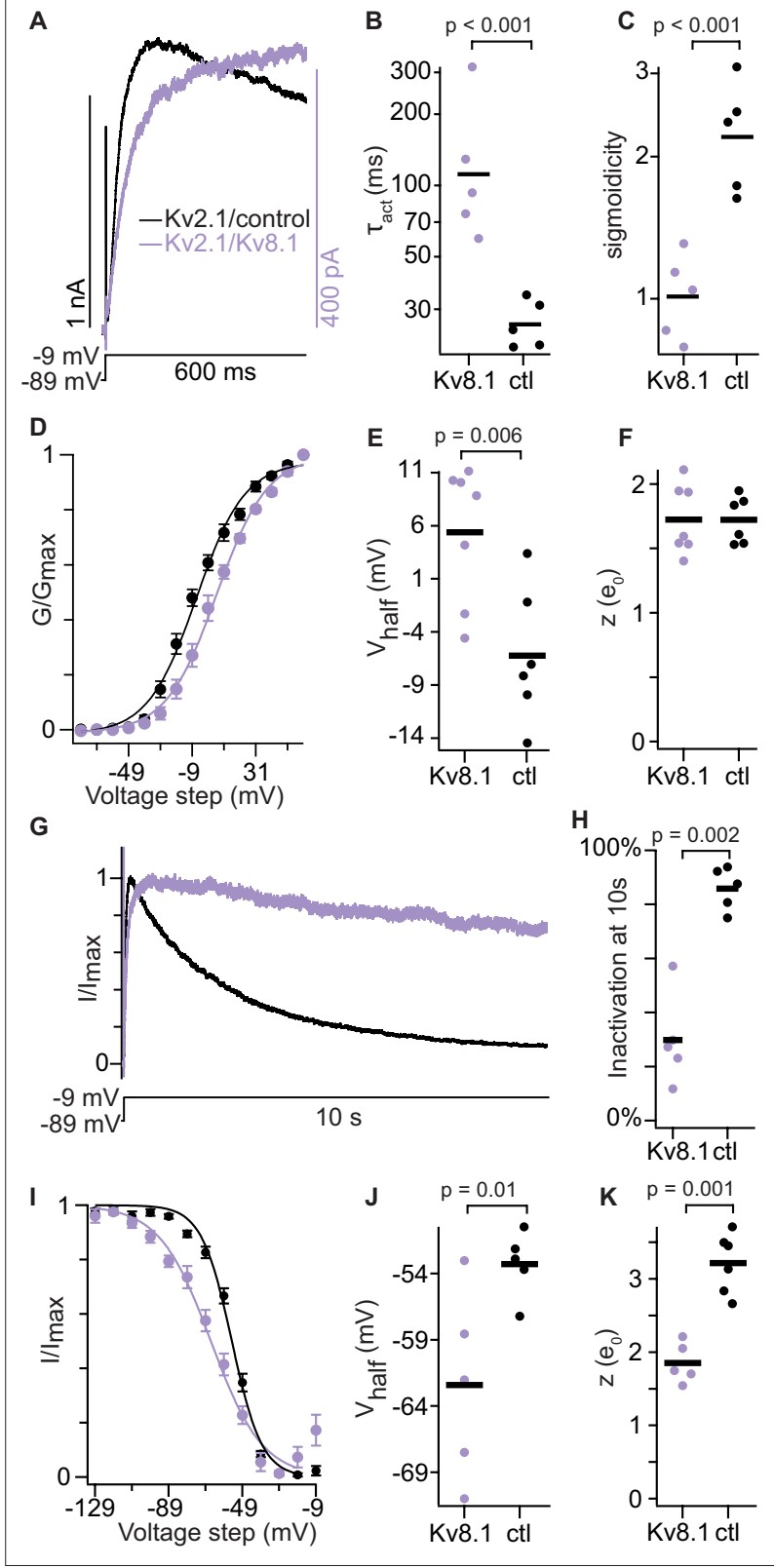

**Figure 4.** RY785-resistant current is consistent with Kv2.1/Kv8.1 heteromers. Kv2.1/8.1 data (purple) are from Kv2.1-CHO cells transfected with Kv8.1, and are in 1 μM RY785. Kv2.1/control (black) were transfected with Navβ2, and are in vehicle control solution. Before measurements, repeated voltage steps to –9 mV were given until currents stabilized. p values are from two-tailed unpaired Wilcoxon rank test. (**A**) Exemplar currents during a step

*Figure 4 continued on next page*

*Figure 4 continued*

to –9 mV. (**B**) Time constants from exponential fit (*Equation 1*). Bars represent geometric mean. (**C**) Sigmoidicity from exponential fit (*Equation 1*). Bars represent geometric mean. (**D**) Conductance-voltage activation relation. Conductance was measured from initial tail currents at –9 mV. Mean ± SEM. Kv2.1/Kv8.1 n=7 cells Kv2.1 n=6 cells. Lines are Boltzmann fits (*Equation 2*) (Kv2.1/Kv8.1: $V_{1/2}$ = 6 ± 1 mV, z=1.6 ± 0.1 $e_0$; Kv2.1/control: $V_{1/2}$ = -6.3 ± 1 mV, z=1.7 ± 0.1 $e_0$). (**E**) Activation $V_{1/2}$ values from individual cells. Bars represent means. (**F**) Activation z values. Bars represent means. (**G**) Exemplar currents during a 10 s step to –9 mV. (**H**) Percent of current inactivated after 10 s at –9 mV. Bars represent means. (**I**) Steady state currents at –9 mV after holding at indicated voltages for 10 s. Normalized to the max and min. Mean ± SEM. Kv2.1/Kv8.1 n=5 cells Kv2.1 n=5 cells. Lines are Boltzmann fits (*Equation 2*) (Kv2.1/Kv8.1: $V_{1/2}$ = –66±1 mV, z=1.8 ± 0.1 $e_0$; Kv2.1/control: $V_{1/2}$ = -54.7 ± 0.8 mV, z=3.1 ± 0.3 $e_0$). (**J**) Inactivation $V_{1/2}$ values from individual cells. Bars represent means. (**K**) Inactivation z values. Bars represent means.

The online version of this article includes the following figure supplement(s) for figure 4:

**Figure supplement 1.** Effect of vehicle control on Kv2.1.

---

conductances (*Liu and Bean, 2014*) yet transcriptomics have revealed little evidence of KvS expression (*Sapio et al., 2020*). We investigated whether SCG neurons have conductances consistent with Kv2/KvS heteromers. As functional characterization alone cannot be trusted to classify their channel mediators of conductances, we define conductances consistent with Kv2/KvS heteromers as 'KvS-like' and conductances consistent with Kv2 homomers as 'Kv2-like'. To help isolate Kv2-like and KvS-like currents, we bathed SCG neurons in a cocktail of Nav, Cav, Kv1, Kv3, and Kv4 inhibitors then recorded voltage-gated currents. We found that exposing SCG neurons to 1 µM RY785 inhibited most of the voltage-gated current, and subsequent addition of 100 nM GxTX to the same neurons inhibited little additional current (*Figure 6A and B*). To quantify inhibition, we analyzed tail currents 10ms after repolarizing to –45 mV. This time window was chosen because a hallmark of Kv2 currents is relatively slow deactivation (*Thorneloe and Nelson, 2003*; *Zheng et al., 2019*). Tail currents after application of 1 µM RY785 were decreased by 88 ± 5% (mean ± SEM) in SCG neurons (*Figure 6B*). Subsequent application of 100 nM GxTX had little further effect. To determine if the RY785-sensitive conductances are consistent with previous reports of Kv2 channels, we examined the biophysical properties of the Kv2-like (RY785-sensitive) currents defined by subtraction (*Figure 6C*). Current activation began to be apparent at –45 mV and had a conductance that was half maximal at –11 mV (*Figure 6D*). The faster

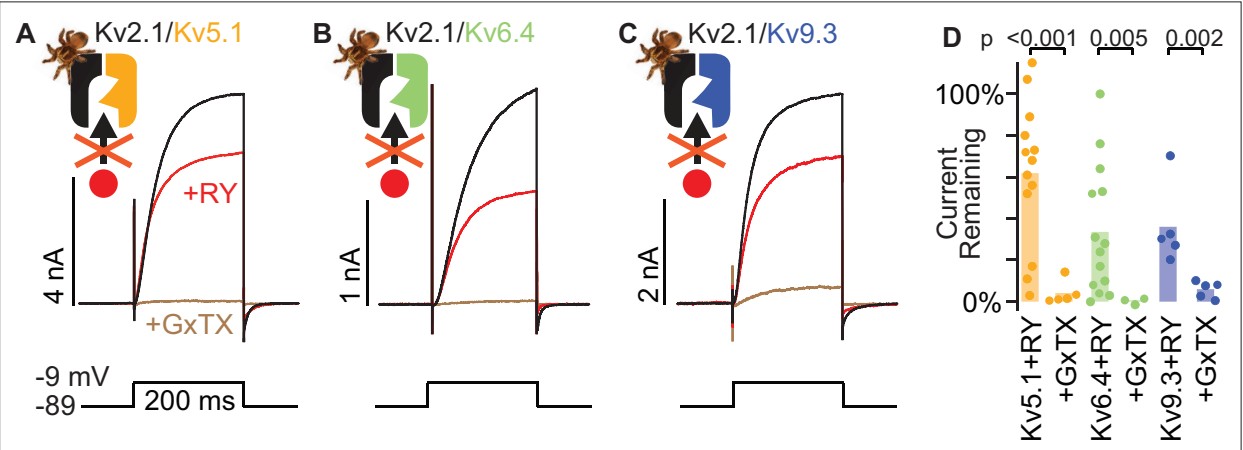

**Figure 5.** A subunit from each KvS subtype is resistant to RY785. Part of the Kv2.1/Kv5.1 dataset was presented previously (*Ferns et al., 2025*). (**A**) Exemplar traces from a voltage-clamped Kv2.1-CHO cell transfected with Kv5.1. Black and red traces are currents before and after application of 1 µM RY785, respectively. Brown trace is current after subsequent application of 1 µM RY785 +100 nM GxTX. (**B**) Exemplar traces from a Kv2.1-CHO cell transfected with Kv6.4. (**C**) Exemplar traces from a Kv2.1-CHO cell transfected with Kv9.3. (**D**) Current remaining after application of 1 µM RY785 or 1 µM RY785 +100 nM GxTX. Bars represent mean. Each point represents current from one cell at the end of a 200ms voltage step to –9 mV. Unpaired Wilcoxon rank tests.

The online version of this article includes the following figure supplement(s) for figure 5:

**Figure supplement 1.** KvS subunits colocalize with Kv2.1 on the surface of CHO cells.

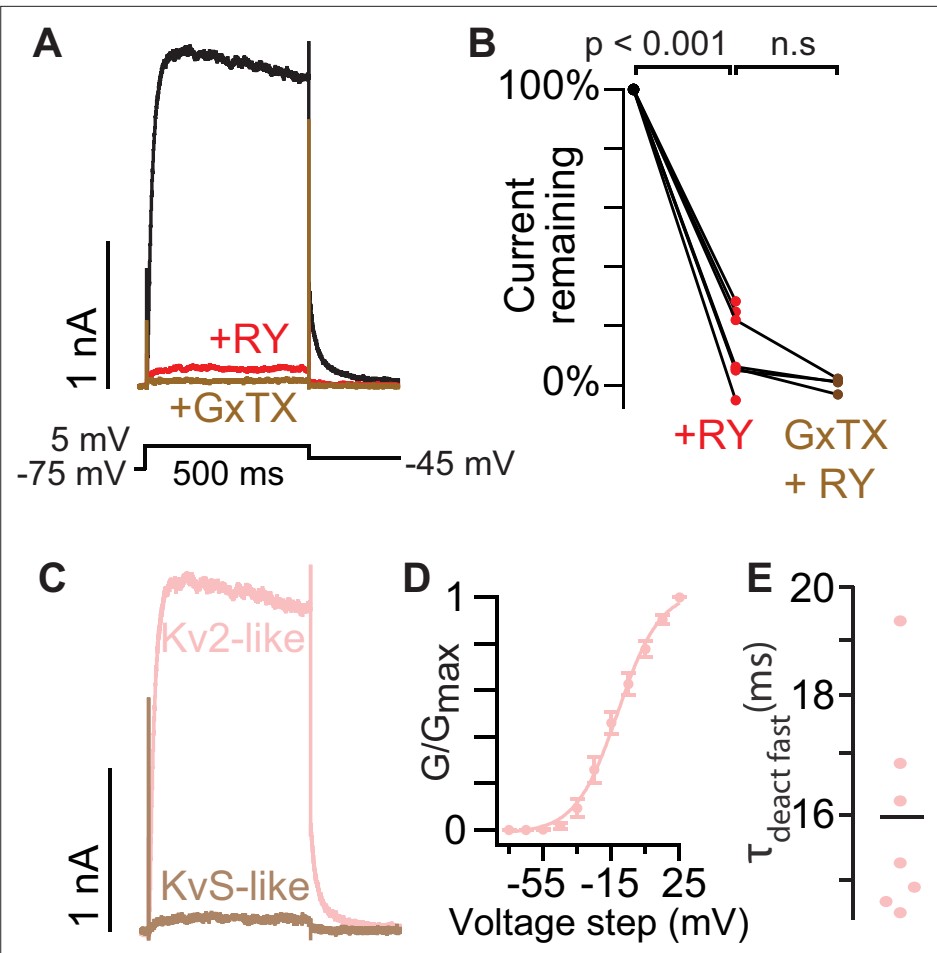

**Figure 6.** The Kv2 conductances of mouse superior cervical ganglion neurons do not have KvS-like pharmacology. (**A**) Exemplar currents from a voltage-clamped SCG neuron. Black and red traces are currents before and after application of 1 μM RY785 respectively. Brown trace is current after subsequent application of 1 μM RY785 +100 nM GxTX. (**B**) Tail current amplitude 10ms after voltage was stepped from +5 mV to -45 mV normalized to current amplitude before RY785. Paired Wilcoxon rank tests, n=7 neurons, N=3 mice. (**C**) Subtracted currents from A. Kv2-like current is the RY785-sensitive current (black trace minus red in A). KvS-like current is the GxTX-sensitive current remaining in RY785 (red trace minus brown in A). (**D**) Conductance-voltage activation relation of Kv2-like current in SCG neurons. Conductance was measured from tail currents at –45 mV. $V_{1/2}$ = –11±1 mV, z=2.1 ± 0.2 $e_0$ Mean ± SEM. n=7 neurons, N=3 mice. (**E**) The faster time constant of a double exponential (***Equation 3***) fit to channel deactivation at –45 mV. Bar represents mean.

component of deactivation of Kv2-like currents in SCG neurons had a time constant of 16ms±0.6 (mean ± SEM) at –45 mV (***Figure 6E***). These results are consistent with reported biophysical properties of Kv2 channels (***Kramer et al., 1998***; ***Liu and Bean, 2014***; ***Tilley et al., 2019***; ***Sepela et al., 2022***). Together these results show that 1 μM RY785 almost completely inhibits endogenous Kv2-like conductances in these mouse neurons, consistent with mouse SCG neurons having few, if any, functional Kv2/KvS heteromers. We cannot rule out that the small amount of current remaining after RY785 (12% of the control) is due to Kv2/KvS heteromers, but its insensitivity to GxTX suggests that it may instead be current from a non-Kv2 channel remaining in the cocktail of K-channel inhibitors.

## The Kv2 conductances of mouse dorsal root ganglion neurons have KvS-like pharmacology

To determine if RY785/GxTX pharmacology could reveal endogenous KvS-containing channels, we next studied neurons likely to express KvS subunits. Mouse dorsal root ganglion (DRG) somatosensory neurons express Kv2 proteins (***Stewart et al., 2024***), have GxTX-sensitive conductances (***Zheng***

*et al., 2019*), and express a variety of KvS transcripts (*Bocksteins et al., 2009*; *Zheng et al., 2019*), yet transcript abundance does not necessarily correlate with functional protein abundance. To record from a consistent subpopulation of mouse somatosensory neurons which has been shown to contain GxTX-sensitive currents and have abundant expression of KvS mRNA transcripts (*Zheng et al., 2019*), we used a *Mrgprd*[GFP] transgenic mouse line which expresses GFP in nonpeptidergic nociceptors (*Zylka et al., 2005*; *Zheng et al., 2019*). Deep sequencing identified that mRNA transcripts for *Kcnf1* (Kv5.1), *Kcng2* (Kv6.2), *Kcng3* (Kv6.3), and *Kcns1* (Kv9.1) are present in GFP[+] neurons of this mouse line (*Zheng et al., 2019*) and we confirmed the presence of *Kcnf1* (Kv5.1) and *Kcns1* (Kv9.1) transcripts in GFP[+] neurons from *Mrgprd*[GFP] mice using RNAscope (*Figure 7—figure supplement 1*). We investigated whether these neurons have conductances consistent with KvS-like pharmacology by performing whole cell voltage clamp on cultured DRG neurons that had clear GFP fluorescence. Voltage-clamped

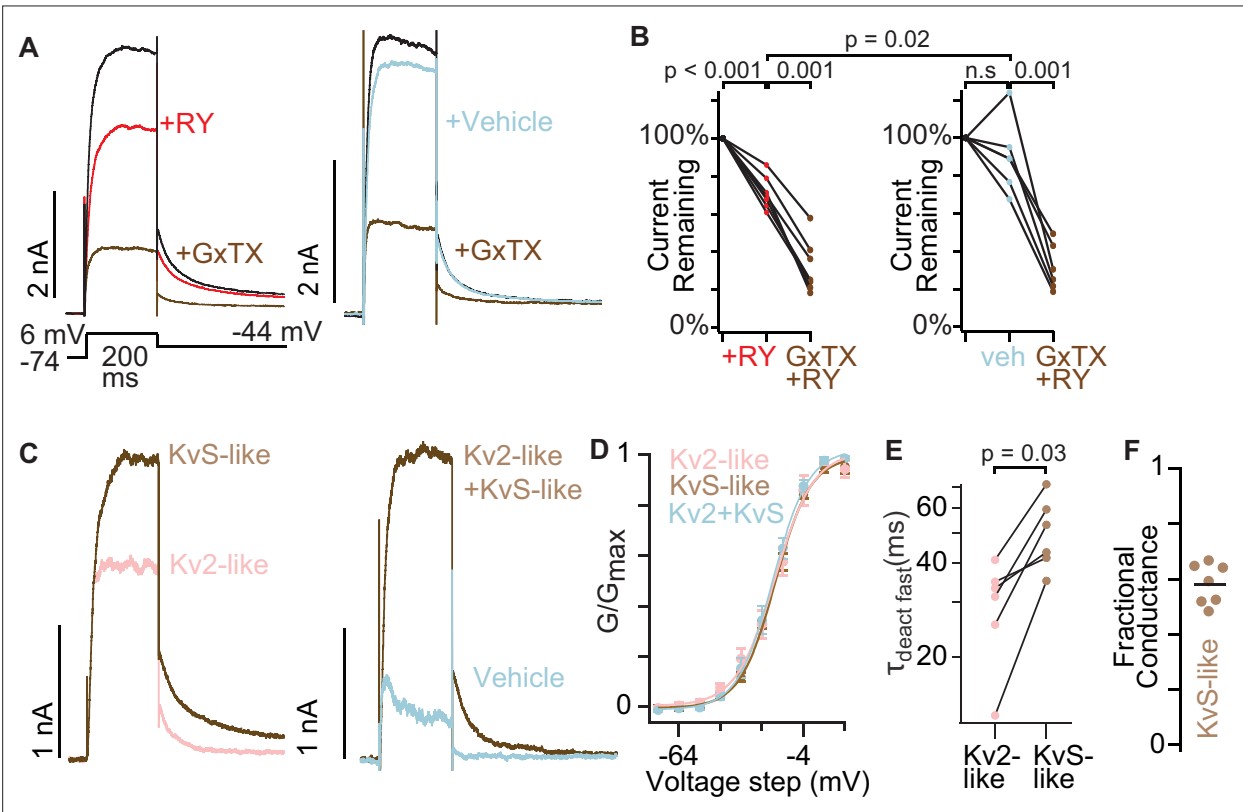

**Figure 7.** The Kv2 conductances of mouse dorsal root ganglion neurons have KvS-like pharmacology. (**A**) Exemplar currents from nonpeptidergic nociceptors, GFP[+] neurons from *Mrgprd*[GFP] mice. (**B**) Tail current amplitude 10ms after voltage was stepped from +6 mV to -44 mV normalized to current amplitude before RY785 or vehicle treatment. Wilcoxon rank tests were paired, except the comparison of RY785 to vehicle which was unpaired. RY785 then GxTX: n=7 neurons, N=4 mice. Vehicle then GxTX: n=6 neurons, N=4 mice. (**C**) Exemplar subtracted currents from A. Kv2-like is the initial current minus RY785 (black trace minus red in A left panel). KvS-like is the current in RY785 minus GxTX (red trace minus brown in A left panel). Kv2-like+KvS like is the current in vehicle minus RY785 +GxTX (blue trace minus brown in A right panel). (**D**) Voltage dependance of activation of subtraction currents in *Mrgprd*[GFP] neurons. Pink points represent Kv2-like currents, brown points represent KvS-like currents, and light blue points represent Kv2 +KvS like currents after vehicle treatment. Conductance was measured from initial tail currents at –44 mV. Kv2-like: $V_{1/2}$ = –18±1 mV, z=2.7 ± 0.3 $e_0$, KvS-like: $V_{1/2}$ = –18±1 mV, z=3 ± 0.2 $e_0$, Kv2 +KvS-like: $V_{1/2}$ = –19±1 mV, z=2.9 ± 0.1 $e_0$. Mean ± SEM. KvS-like and Kv2-like n=7 neurons N=4 mice, Kv2 +KvS like n=6 neurons N=4 mice. (**E**) The faster time constant of a double exponential fit (*Equation 3*) to channel deactivation at –44 mV. p value represents paired Wilcoxon rank test. (**F**) Fractional KvS-like conductance relative to the total RY785 +GxTX-sensitive conductance. KvS-like is only sensitive to GxTX. Bar represents mean.

The online version of this article includes the following figure supplement(s) for figure 7:

**Figure supplement 1.** Nonpeptidergic nociceptors express *Kcnf1* (Kv5.1) and *Kcns1* (Kv9.1) mRNA transcripts.

**Figure supplement 2.** RY785 resistant currents from Kv2.1-CHO cells transfected with Kv5.1 or Kv9.1 deactivate slower than currents from untransfected Kv2.1-CHO cell.

**Figure supplement 3.** The Kv2 conductances of mouse dorsal root ganglion neurons have KvS-like pharmacology in the absence of the cocktail of inhibitors.

neurons were bathed in the same cocktail of channel inhibitors used on SCG neurons, plus the Nav1.8 inhibitor A-803467. Application of 1 µM RY785 inhibited outward currents somewhat, but unlike in SCG neurons, a prominent delayed-rectifier outward conductance with slow deactivation remained (*Figure 7A* left panel). Tail currents in 1 µM RY785 decreased 29 ± 3% (mean ± SEM) (*Figure 7B* left panel). Subsequent application of 100 nM GxTX decreased tail currents by 68 ± 5% (mean ± SEM) of their original amplitude before RY785. We do not know the identity of the outward current that remains in the cocktail of inhibitors + RY785+GxTX. We observed variable current run-up or run-down but no significant effect of vehicle in blinded, interleaved experiments, while RY785 significantly decreased tail currents relative to vehicle controls (*Figure 7A and B* right panel). We do not know what conductances the vehicle solution exchange affects, the changes appear to be time-dependent or due to the solution exchange itself. Concurrent application of 100 nM GxTX and 1 µM RY785 to neurons in vehicle decreased currents by 69 ± 5% (mean ± SEM).

To determine if the RY785- and GxTX-sensitive conductances in GFP+ neurons from *Mrgprd*GFP mice are consistent with previous reports of Kv2 homomeric or Kv2/KvS heteromeric channels, we examined the biophysical properties of the Kv2-like (RY785-sensitive) and KvS-like (RY785-resistant, GxTX-sensitive) currents defined by subtraction (*Figure 7C*). Obvious Kv2-like and KvS-like channel conductances began at –44 mV and had half-maximal conductances around –19 mV (*Figure 7D*), consistent with Kv2 and many KvS-containing channels (*Kramer et al., 1998*; *Richardson and Kaczmarek, 2000*; *Sano et al., 2002*; *Thorneloe and Nelson, 2003*). While changes in inactivation are prominent with KvS subunits, we did not investigate inactivation in neurons because the lengthy depolarizations required often resulted in irreversible leak current increases that degraded the accuracy of RY785/GxTX subtraction current quantification. We did note that the KvS-like currents deactivated slower than Kv2-like currents (*Figure 7E*), consistent with the effects of several KvS subunits whose transcripts are expressed in nociceptor DRG neurons. Kv5.1, Kv6.3, and Kv9.1 all slow deactivation of Kv2 conductances in heterologous cells (*Salinas et al., 1997b*; *Kramer et al., 1998*; *Sano et al., 2002*), and in Kv2.1-CHO cells transfected with Kv5.1, we confirmed that RY785-resistant currents deactivate slower than Kv2.1 controls (*Figure 7—figure supplement 2*). Together, these results indicate that, in these mouse DRG neurons, RY785-sensitive currents are Kv2-like, while RY785-resistant yet GxTX-sensitive currents are KvS-like. Under these conditions, 58 ± 3% (mean ± SEM) of the delayed rectifier conductance was resistant to RY785 yet sensitive to GxTX (KvS-like; *Figure 7F*). We note that the ratio of KvS- to Kv2-like conductances is expected to vary with holding potential, as KvS subunits can change the degree and voltage-dependence of steady state inactivation (e.g. *Figure 4I*).

We also tested the other major mouse C-fiber nociceptor population, peptidergic nociceptors, to determine if this subpopulation also has conductances resistant to RY785 yet sensitive to GxTX. We voltage clamped DRG neurons from a *Calca*GFP mouse line that expresses GFP in peptidergic nociceptors (*Gong et al., 2003*). Deep sequencing has identified mRNA transcripts for *Kcng2* (Kv6.2), *Kcng3* (Kv6.3), *Kcnv1* (Kv8.1) and *Kcns3* (Kv9.3) present in GFP+ neurons, an overlapping but distinct set of KvS subunits from the *Mrgprd*GFP non-peptidergic population (*Zheng et al., 2019*). In GFP+ neurons from *Calca*GFP mice, we found that a fraction of outward current was inhibited by 1 µM RY785 and additional current inhibited by 100 nM GxTX (*Figure 7—figure supplement 3A-C*). In these experiments, 58 ± 2% (mean ± SEM) was KvS-like (*Figure 7—figure supplement 3D*) identifying that KvS-like conductances are present in these peptidergic nociceptors. For *Calca*GFP neurons we did not include the Kv1, Kv3, Kv4, Nav, and Cav channel inhibitor cocktail used for other neuron experiments, indicating that the cocktail of inhibitors is not required to identify KvS-like conductances. Overall, these results show that the unique pharmacology of RY785 and GxTX can reveal endogenous KvS-like conductances in the major subtypes of mouse C-fiber nociceptors.

## The Kv2 conductances of human dorsal root ganglion neurons have KvS-like pharmacology

Human DRG neurons express Kv2 proteins (*Stewart et al., 2024*), and express KvS transcripts (*Ray et al., 2018*) suggesting that they may have Kv2/KvS conductances. We performed whole-cell voltage clamp on cultured human DRG neurons, choosing smaller-diameter neurons and using the same solutions as *Mrgprd*GFP mouse DRG neuron recordings. In human DRG neurons, a fraction of outward current was inhibited by 1 µM RY785 and additional current was inhibited by 100 nM GxTX, consistent with the presence of KvS-like conductances (*Figure 8A, B and C*). These RY785- or GxTX-sensitive

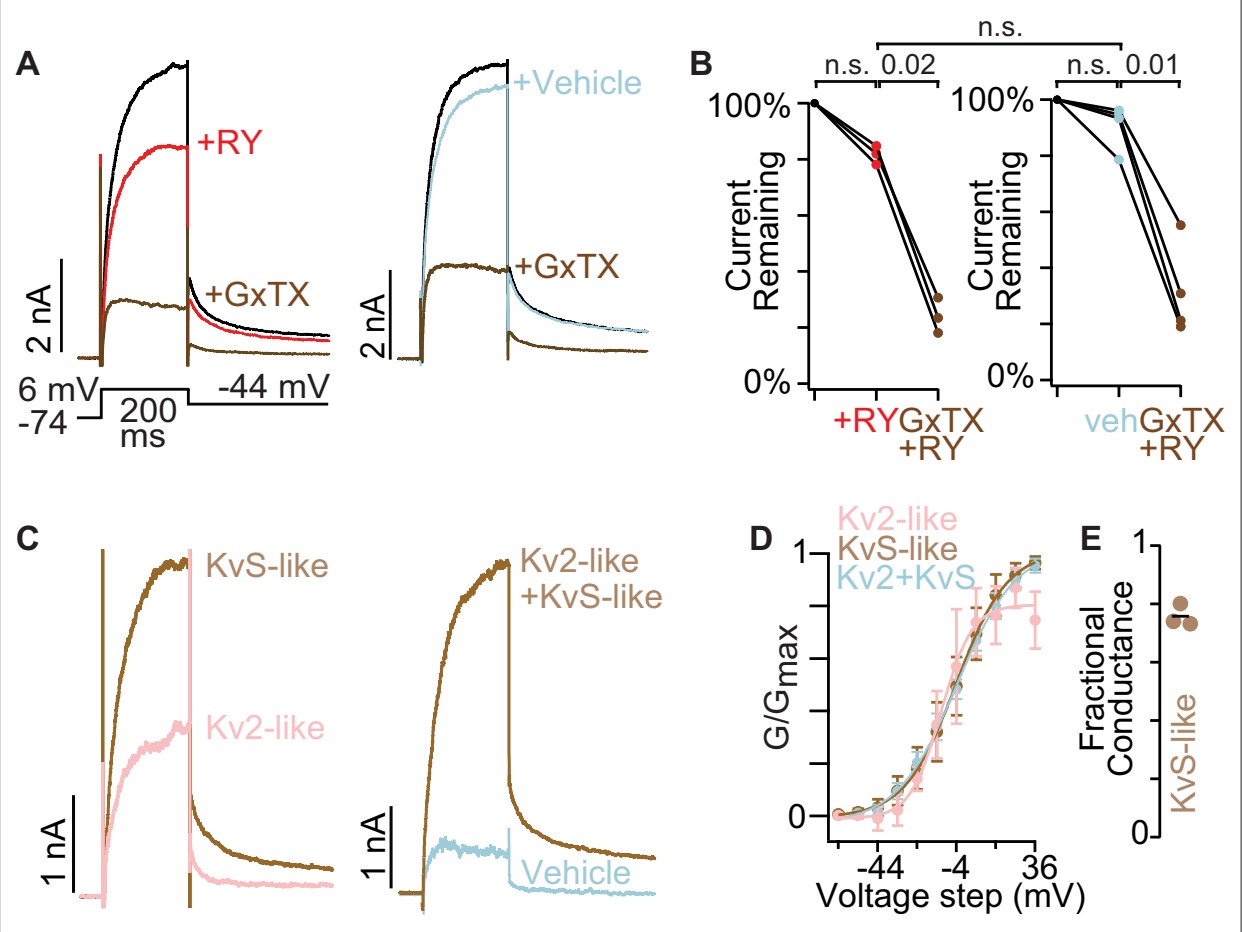

**Figure 8.** The Kv2 conductances of human dorsal root ganglion neurons have KvS-like pharmacology. (**A**) Exemplar currents from human dorsal root ganglion neurons. (**B**) Tail current amplitude 10ms after voltage was stepped from +6 mV to -44 mV normalized to current amplitude before RY785 or vehicle treatment. Wilcoxon rank tests were paired. RY785 then GxTX: n=3 neurons. Vehicle then RY785 and GxTX: n=4 neurons. All neurons from same human. (**C**) Exemplar subtracted currents from A. Kv2-like is the initial current minus RY785 (black trace minus red in A left panel). KvS-like is the current in RY785 minus GxTX (red trace minus brown in A left panel). Kv2 +KvS like is the current in vehicle minus RY785 +GxTX (blue trace minus brown in A right panel). (**D**) Voltage dependence of activation of subtraction currents in human dorsal root ganglion neurons. Pink points represent Kv2-like currents, brown points represent KvS-like currents, and blue points represent Kv2 +KvS like currents after vehicle treatment. Conductance was measured from initial tail currents at –44 mV. Mean ± SEM. Kv2 +KvS like n=3 neurons N=1 human, KvS +Kv2 like n=4 neurons N=1 human. (**E**) Fractional KvS-like conductance relative to the total RY785 +GxTX-sensitive conductance. KvS-like is only sensitive to GxTX. Bar represents mean.

conductances became apparent near –44 mV and were half-maximal between –14 and –4 mV (**Figure 8D**). Of the total conductance sensitive to RY785 +GxTX in these human DRG neurons, 76 ± 2% (mean ± SEM) was KvS-like (**Figure 8E**). Unlike mouse neurons, we did not detect a significant difference in tail currents of RY785 versus vehicle controls. However, RY785-subtracted currents always had Kv2-like biophysical properties whereas vehicle-subtraction currents had variable properties that precluded the same biophysical analysis. Overall, these results show that human DRG neurons can produce endogenous voltage-gated currents with pharmacology and gating consistent with Kv2/KvS heteromeric channels.

## Discussion

These results identify a method for pharmacologically isolating conductances of Kv2/KvS heteromers and Kv2-only channels. RY785 blocks homomeric Kv2 channels, and subsequent application of GxTX selectively inhibits Kv2/KvS heteromeric channels. Such a protocol can aid in identification of Kv2/KvS conductances separately from the Kv2 homomer conductances that are likely to be in the same cell. Characterization of these now separable conductances can reveal impacts of Kv2 channels and

KvS-containing channels on electrophysiological signaling. This is valuable as there are few other tools to probe the contributions of KvS subunits to electrical signaling in native cells and tissues.

It is remarkable that resistance to RY785 is shared across all of the KvS subtypes. We found Kv2.1/Kv8.1 conductances to be ~1000 times less sensitive to RY785 than Kv2.1 homomer conductances in the same cell line. Based on the observation that >50% of Kv current was resistant to 1 μM RY785 in some Kv2.1-CHO cells transfected with Kv5.1, Kv6.4, or Kv9.3, these Kv2.1/KvS channels are expected

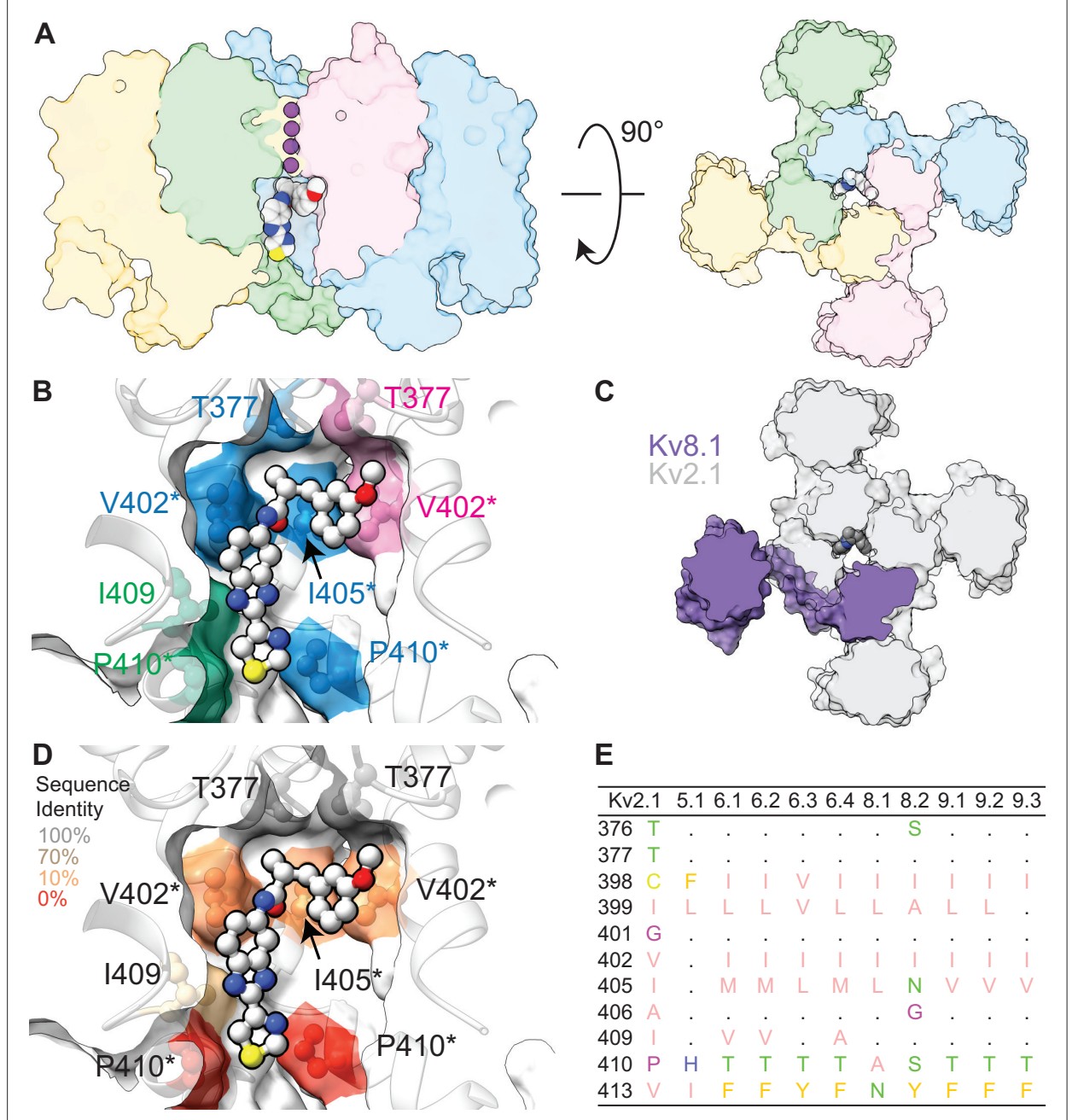

**Figure 9.** Structural models of RY785 docked to a Kv2.1 homomer and 3:1 Kv2.1:Kv8.1 heteromer. (**A**) Model of RY785 docked to a Kv2.1 homomer (PDB 8SD3). Left: transmembrane view. Right: Extracellular view. Individual subunits colored distinctly. RY785 represented by the Corey–Pauling–Koltun space-filling model and coloring scheme. Potassium (purple spheres) from PDB 8SD3 are shown in selectivity filter. (**B**) A zoom in of central cavity in transmembrane view. Colors indicate subunit of residues making Van der Waals contacts with RY785. Human Kv2.1 sequence numbering. Asterisks indicate residues that vary in Kv8.1. (**C**) Model of RY785 docked to a 3:1 Kv2.1:Kv8.1 heteromer. RY785 makes no contact with Kv8.1. (**D**) A replica of panel B with heatmap coloring to indicate sequence conservation of Kv2.1 to KvS subunits. (**E**) Sequence comparison of pore-lining S6 residues of Kv2.1 and all human KvS subunits. Zappo coloring indicates physiochemical identity. A dot indicates sequence conservation with Kv2.1.

to have $IC_{50}$ >1 µM, at least 100-fold less sensitive than the ~6 nM $IC_{50}$ for Kv2.1 in this cell line. This suggests that the RY785 inhibitory site is substantially disrupted by KvS subunits. We analyzed computational structural models of RY785 docked to a Kv2.1 homomer and a 3:1 Kv2.1:Kv8.1 heteromer (*Figure 9*) to gain structural insight into how KvS subunits might interfere with RY785 binding. We used Rosetta to dock RY785 to a cryo-EM structure of a Kv2.1 homomer in an apparently open state (*Fernández-Mariño et al., 2023*). The top-scoring docking pose has RY785 positioned below the selectivity filter and off-axis of the pore (*Figure 9A*), similar to a stable pose observed in molecular dynamics simulations (*Zhang et al., 2024*). In this pose, RY785 contacts a collection of Kv2.1 residues that vary in every KvS subtype (*Figure 9B, D and E*). Notably, RY785 bound similarly to a 3:1 model of Kv2.1/Kv8.1, in contact with the three Kv2.1 subunits, yet avoided the Kv8.1 subunit (*Figure 9C*). This is consistent with RY785 binding less well to Kv2.1/Kv8.1 heteromers, and also suggests that a 3:1 Kv2:KvS channel could retain an RY785 binding site when open. However, the RY785 resistance of Kv2/KvS heteromers may primarily arise from perturbed interactions with the constricted central cavity of closed channels. In homomeric Kv2.1, RY785 becomes trapped in closed channels and prevents their voltage sensors from fully activating, indicating that RY785 must interact differently with closed channels (*Marquis and Sack, 2022*). Here we found that Kv2.1/Kv8.1 current rapidly recovers following washout of RY785, suggesting that Kv2.1/Kv8.1 heteromers do not readily trap RY785 (*Figure 2—figure supplement 1*). Overall, the structural modeling suggests that KvS subunits sterically interfere with RY785 binding to the central cavity, while functional data suggest KvS subunits disrupt RY785 trapping in closed states.

While we have identified a potential means to isolate conductances from Kv2 channels containing KvS subunits, it is important to consider the limitations of these findings:

First, although every KvS subunit we tested makes conductance resistant to RY785, including Kv5.1 the most similar to Kv2.1 (*Jegla et al., 2024*; *Simonson et al., 2025*) and spanning all the KvS subtypes (Kv5, Kv6, Kv8, and Kv9), we have not tested all KvS subunits, species variants, cell types, or voltage regimens. Any of these could alter the RY785 $IC_{50}$. Pharmacology can yield surprises, such as the unexpected resistance of human Nav1.7 to saxitoxin (*Walker et al., 2012*). The degree of resistance to RY785 may vary among KvS subunits as with other central cavity drugs such as tetraethylammonium and 4-aminopyridine (*Post et al., 1996*; *Thorneloe and Nelson, 2003*; *Stas et al., 2015*). Also, the degree of resistance to RY785 may vary if Kv2:KvS subunit stoichiometry varies. With high doses of RY785, we found that the concentration-response characteristics of Kv2.1/Kv8.1 in CHO cells revealed hallmarks of a homogenous channel population with a Hill slope close to 1 (*Figure 2B*). However, other KvS subunits might assemble in multiple stoichiometries and result in pharmacologically-distinct heteromer populations.

Second, it is possible that RY785 can modulate other voltage-gated channels. However, we think RY785 is unlikely to have substantial off-target effects, as RY785 is much less potent against other non-Kv2 voltage-gated potassium channel subfamilies as well as voltage-gated sodium and calcium channels (*Herrington et al., 2011*).

Third, concentrations of RY785 which partially blocked Kv2.1/Kv8.1 modified the gating of the voltage-activated conductance, indicating that RY785 can alter properties of heteromer conductance. Also, channel state-dependent binding is expected to influence the affinity of RY785. These effects could create complications in analyzing current subtractions.

Fourth, these are functional characterizations of currents and should not be solely trusted to classify the molecular identity of the underlying channels. While it seems unlikely that pore-forming subunits other than KvS subunits complex with Kv2 subunits, other factors could potentially disrupt the RY785 pharmacology as KvS subunits do. Regulation of Kv2 channel gating could potentially allosterically disrupt RY785 inhibition. Extensive homeostatic regulation of Kv2.1 gating maintains neuronal excitability *Misonou et al., 2006*; for example, ischemia (*Misonou et al., 2005*; *Aras et al., 2009*), glutamate (*Misonou et al., 2008*), phosphorylation (*Murakoshi et al., 1997*), SUMOylation (*Plant et al., 2011*), and AMIGO auxiliary subunits (*Peltola et al., 2011*; *Maverick et al., 2021*) all alter Kv2 gating. Intriguingly, some KvS subunits are reported to functionally interact with Kv7 subunits (*Renigunta et al., 2024*). Also, the cocktail of inhibitors used in most neuron experiments here could potentially alter RY785 or GxTX action against KvS/Kv2 channels.

Despite these possibilities, we think the most parsimonious interpretation of RY785-resistant, GxTX-sensitive conductances is that they are produced by heteromeric Kv2/KvS channels. However, it

is important to consider these other possibilities when interpreting RY785 resistance of GxTX-sensitive conductances. With these caveats in mind, we suggest that RY785-resistance combined with GxTX-sensitivity is strong evidence for heteromeric Kv2/KvS currents.

Delayed-rectifier Kv2-like conductances are prominent in many electrically excitable cell types. While we found only RY785-sensitive Kv2-like conductances in SCG neurons, Kv2/KvS heteromer-like conductances were dominant in DRG neurons. This striking contrast is consistent with KvS transcript abundances (*Zheng et al., 2019*; *Sapio et al., 2020*). Beyond these cell types, it is unclear how prevalent Kv2/KvS heteromer conductances are. While Kv2 subunits are broadly expressed in electrically excitable cells throughout the brain and body, transcripts for KvS subunits have unique expression patterns that are specific to each KvS subtype (*Bocksteins, 2016*) and can fluctuate with age (*Regnier et al., 2016*). However, transcript levels alone are not sufficient to predict protein levels (*Liu et al., 2016*). We found recently that native Kv2 channels in mouse brain contain KvS subunit proteins, including Kv5.1, Kv8.1, Kv9.1, and Kv9.2. Notably, the KvS mass spectral abundance relative to Kv2.1 ranged from ≈18% for Kv5.1–2% for Kv9.1 (*Ferns et al., 2025*), and Kv5.1 protein expression was largely restricted to cortical neurons. Thus, it seems likely that significant Kv2/KvS heteromeric conductances also exist in specific subsets of brain neurons.

The physiological role of Kv2/KvS heteromers in neurons and other excitable cells remains enigmatic. Previous studies have used knockout mice (*Regnier et al., 2017*; *Miyamae et al., 2021*), transient transfection of KvS subunits (*Lee et al., 2020*), siRNA knockdown (*Tsantoulas et al., 2012*), and modeling (*Miyamae et al., 2021*) to probe the presence of endogenous and functional KvS-containing channels, and these methods can identify phenotypic changes that suggest potential roles of KvS-containing channels. Application of GxTX in vivo could be used to probe the physiological roles of Kv2 with or without KvS subunits, while RY785 selectively targets Kv2-only channels. Genetic mutations and gene targeting studies have linked disruptions in the function of KvS-containing channels to epilepsy (*Jorge et al., 2011*), neuropathic pain sensitivity (*Tsantoulas et al., 2018*), labor pain (*Lee et al., 2020*) and retinal cone dystrophy (*Wu et al., 2006*; *Hart et al., 2019*; *Inamdar et al., 2022*), stressing their functional importance in specific cell types. The unique pharmacology of KvS-containing channels identified here provides a new and direct method of identifying conductances mediated by KvS-containing channels in native neurons and establishing what contributions KvS subunits make to electrophysiological signaling.

Finally, these findings also support the potential utility of KvS channels as drug targets. Kv2-targeted drug leads have poor tissue and cell specificity and suffer from pronounced side effects (*Li et al., 2013*). KvS transcripts show far greater tissue- and cell-type specific expression relative to Kv2 (*Bishop et al., 2015*; *Bocksteins, 2016*), and we identified prominent Kv2 conductances with KvS-like pharmacology in mouse nociceptor and human DRG neurons. Consequently, KvS-targeted drugs could offer greater specificity and the ability to modulate neuronal excitability in a variety of pathological contexts, such as neuropathic pain.

## Materials and methods

### Human tissue collection

Human dorsal root ganglia (DRG) were obtained from Sierra Donor Services. The donor was a 58-year-old Asian Indian female and DRG were from the 1st and 2nd lumbar region (cause of death: Stroke). DRG were extracted 6 hr after aortic cross clamp and placed in an ice cold N-methyl-D-glucamine-artificial cerebral spinal fluid (NMDG-aCSF) solution containing in mM: 93 NMDG, 2.5 KCl, 1.25 $NaH_2PO_4$, 30 $NaHCO_3$, 20 HEPES, 25 Glucose, 5 L-Ascorbic acid, 2 Thiourea, 3 Na pyruvate, 10 $MgSO_4$ and 0.5 $CaCl_2$ pH adjusted to 7.4 with HCl. Human DRG were obtained from the organ donor with full legal consent for use of tissue for research in compliance with procedures approved by Sierra Donor Services.

### Chinese hamster ovary (CHO) cell culture and transfection

The CHO-K1 cell line transfected with a tetracycline-inducible rat Kv2.1 construct (Kv2.1-CHO; *Trapani and Korn, 2003*) was cultured as described previously (*Tilley et al., 2014*). Transfections were achieved with Lipofectamine 3000 (Life Technologies, L3000001). 1 µl Lipofectamine was diluted, mixed, and incubated in 25 µl of Opti-MEM (Gibco, 31985062). Concurrently, 0.5 µg of KvS

or AMIGO1 or Navβ2, 0.5 µg of pEGFP, 2 µl of P3000 reagent and 25 µl of Opti-MEM were mixed. DNA and Lipofectamine 3000 mixtures were mixed and incubated at room temperature for 15 min. This transfection cocktail was added to 1 ml of culture media in a 24-well cell culture dish containing Kv2.1-CHO cells and incubated at 37 °C in 5% $CO_2$ for 6 hr before the media was replaced. Immediately after media was replaced, Kv2.1 expression was induced in Kv2.1-CHO cells with 1 µg/ml minocycline (Enzo Life Sciences, ALX-380–109 M050), prepared in 70% ethanol at 2 mg/ml. Voltage clamp recordings were performed 12–24 hr later. We note that the expression method of Kv2/KvS heteromers used here is distinct from previous studies which show that the KvS:Kv2 mRNA ratio can affect the expression of functional Kv2/KvS heteromers (*Salinas et al., 1997b*; *Pisupati et al., 2018*). We validated the functional Kv2/KvS heteromer expression using voltage clamp to establish distinct channel kinetics and the presence of RY785-resistant conductance in KvS-transfected cells and using immunohistochemistry to label apparent surface localization of KvS subunits (*Figure 1*, *Figure 5—figure supplement 1*, *Figures 4 and 5*). During recordings, the experimenter was blinded as to whether cells had been transfected with KvS, or Navβ2 or AMIGO1. Human Kv5.1, human Kv6.4 and human Kv8.1, AMIGO1-YFP, and pEGFP plasmids were gifts from James Trimmer (University of California, Davis, Davis, CA). Human Kv9.1 and human Kv9.3 plasmids were purchased from Addgene. Human Navβ2 plasmid was a kind gift from Dr. Alfred George (*Lossin et al., 2002*).

## Neuron cell culture

### Mouse

Studies were approved by the UC Davis and Harvard Medical School Institutional Animal Care and Use Committees and conform to guidelines established by the NIH. Mice were maintained on a 12 hr light/dark cycle, and food and water were provided ad libitum. The *Mrgprd*[GFP] (MGI: 3521853) and *Calca*[GFP] (MGI: 2151253) mouse lines were a generous gift from David Ginty at Harvard.

Cervical, thoracic and lumbar dorsal root ganglia (DRGs) were harvested from 7- to 10-week-old *Mrgprd*[GFP] or 2- to 4-week-old *Calca*[GFP] mice and transferred to Hank's buffered saline solution (HBSS) (Invitrogen). Ganglia were treated with collagenase (2 mg/ml; Type P, Sigma-Aldrich) in HBSS for 15 min at 37 °C followed by 0.05% Trypsin-EDTA (Gibco) for 2.5 min with gentle rotation. Trypsin was neutralized with culture media (MEM, with l-glutamine, Phenol Red, without sodium pyruvate) supplemented with 10% horse serum (heat-inactivated; Gibco), 10 U/ml penicillin, 10 µg/ml streptomycin, MEM vitamin solution (Gibco), and B-27 supplement (Gibco). Serum-containing media was decanted and cells were triturated using a fire-polished Pasteur pipette in MEM culture media containing the supplements listed above. Cells were plated on laminin-treated (0.05 mg/ml, Sigma-Aldrich) 5 mm German glass coverslips (Bellco Glass, 1943–00005), which had previously been washed in 70% ethanol and sterilized with ultraviolet light. Cells were then incubated at 37 °C in 5% $CO_2$. Cells were used for electrophysiological experiments 24–38 hr after plating.

Superior cervical ganglia (SCG) were harvested from Swiss Webster (CFW) mice (postnatal day 13–15, either sex) and treated for 20 min at room temperature (RT) with 20 U/ml papain (Worthington Biochemical), 5 mM dl-cysteine, 1.25 mM EDTA, and 67 µM β-mercaptoethanol in a $Ca^{2+}$, $Mg^{2+}$-free (CMF) Hank's solution (Gibco) supplemented with 1 mM Sodium Pyruvate (Sigma-Aldrich, St. Louis, MO), and 5 mM HEPES (Sigma-Aldrich, St. Louis, MO). Ganglia were then treated for 20 min at 37 °C with 3 mg/ml collagenase (type I; Roche Diagnostics) and 3 mg/ml dispase II (Roche Diagnostics) in CMF Hank's solution. Cells were dispersed by trituration with fire-polished Pasteur pipettes in a solution composed of two media combined in a 1:1 ratio: Leibovitz's L-15 (Invitrogen) supplemented with 5 mM HEPES, and DMEM/F12 medium (Invitrogen). Cells were then plated on glass coverslips and incubated at 37 °C (95% $O_2$, 5% $CO_2$) for 1 hr, after which Neurobasal medium (Invitrogen) with B-27 supplement (Invitrogen), penicillin and streptomycin (Sigma) was added to the dish. Cells were incubated at 25 °C (95% $O_2$, 5% $CO_2$) and used within 10 hr.

### Human

Dura were removed from human DRG with a scalpel in ice cold NMDG-aCSF solution (*Valtcheva et al., 2016*). Human DRG were then cut into approximately 1-mm-thick sections and were placed in 1.7 mg/mL Stemxyme (Worthington Biochemical, LS004107) and 6.7 mg/mL DNAse I (Worthington Biochemical, LSOO2139) diluted in HBSS (Thermo Fisher Scientific, 14170161) for 12 hr at 37 °C. DRG were then triturated with a fire-polished Pasteur pipette and passed through a 100 µm cell strainer.

Cells were then spun at 900 × g through 10% BSA. The supernatant was removed, and cells were resuspended in human DRG culturing media that contained 1% penicillin/streptomycin, 1% GlutaMAX (Gibco, 35050–061), 2% NeuroCult SM1 (05711, Stemcell technologies), 1% N2 Supplement (Thermo Fisher Scientific, 17502048), 2% FBS (Gibco, 26140–079) diluted in BrainPhys media (Stemcell tehnologies, 05790). DRG neurons were plated on poly-D-lysine treated (0.01 mg/mL) 5 mm German glass coverslips, which had previously been washed in 70% ethanol and sterilized with ultraviolet light. DRG neurons were then incubated at 37 °C in 5% $CO_2$. Human DRG neuron experiments were performed up to 7 days after plating.

## Whole cell voltage clamp of CHO cells

Voltage clamp was achieved with a dPatch amplifier (Sutter Instruments) run by SutterPatch software (Sutter Instruments). Solutions for Kv2.1-CHO cell voltage-clamp recordings: CHO-internal (in mM) 120 K-methylsulfonate, 10 KCl, 10 NaCl, 5 EGTA, 0.5 $CaCl_2$, 10 HEPES, 2.5 MgATP pH adjusted to 7.2 with KOH, 289 mOsm. CHO-external (in mM) 145 NaCl, 5 KCl, 2 $CaCl_2$, 2 $MgCl_2$, 10 HEPES pH adjusted to 7.3 with NaOH, 298 mOsm. Osmolality was measured with a vapor pressure osmometer (Wescor, 5520). The liquid junction potential of –9 mV between these solutions was accounted for. The liquid junction potential was calculated according to the stationary Nernst–Planck equation (*Marino.M and Brogioli, 2014*) using LJPcalc (RRID:SCR_025044). For voltage-clamp recordings, Kv2.1-CHO cells were detached in a PBS-EDTA solution (Gibco, 15040–066), spun at 500 × g for 2 min and then resuspended in 50% cell culture media and 50% CHO-external recording solution. Cells were then added to a recording chamber (Warner, 64–0381) and were rinsed with the CHO-external patching solution after adhering to the bottom of the recording chamber. Transfected Kv2.1-CHO cells were identified by GFP fluorescence and were selected for whole cell voltage clamp. Thin-wall borosilicate glass recording pipettes (Sutter, BF150-110-10) were pulled with blunt tips, coated with silicone elastomer (Sylgard 184, Dow Corning), heat cured, and tip fire-polished to resistances less than 4 MΩ. Series resistance of 2–14 MΩ was estimated from the Sutterpatch whole-cell parameters routine. Series resistance compensation between 13 and 90% was used to constrain voltage error to less than 15 mV; compensation feedback lag was 6 µs for most experiments or 100 µs for concentration-effect experiments. Capacitance and ohmic leak were subtracted using a P/4 protocol. Output was low-pass filtered at 5 kHz using the amplifier's built-in Bessel and digitized at 25 kHz or, for concentration-effect experiments, 1 and 10 kHz. Experiments were performed on Kv2.1-CHO cells with membrane resistance greater than 1 GΩ assessed prior to running voltage clamp protocols while cells were held at a membrane potential of –89 mV. RY785 (gift from Bruce Bean, Harvard, or Cayman, 19813) was prepared in DMSO as a 1 mM stock for dilutions to 1 µM or a 35 mM stock for concentration-effect experiments. Stocks of GxTX-1E Met35Nle (*Tilley et al., 2014*) in water were 10 µM. Stocks were stored frozen and diluted in recording solution just prior to application to cells. Solutions were flushed over cells at a rate of approximately 1 ml/min. Concentrated RY785 and GxTX stocks were stored at –20 °C. Kv2.1-CHO cells were given voltage steps from –89 mV to –9 mV for 200ms every 6 s during application of RY785 until currents stabilized. When vehicle control was applied to cells, –9 mV steps were given for a similar duration. All RY785 solutions contained 0.1% DMSO. Vehicle control solutions also contained 0.1% DMSO but lacked RY785. Perfusion lines were cleaned with 70% ethanol then doubly-deionized water. For concentration-effect experiments, changes in current amplitude due to solution exchange were controlled for by treating every other tested cell with multiple washes of the same, 0.35 µM RY785 solution instead of increasing concentrations of RY785. The timing and duration of these control washes was similar to that of the washes in concentration-effect experiments.

## Whole cell voltage clamp of mouse and human dorsal root ganglion neurons

Whole cell recordings from *Mrgprd*[GFP] mouse and human neurons were performed using the same methods as CHO cell recordings with the following exceptions. Voltage clamp was achieved with a dPatch amplifier run by SutterPatch software or an AxoPatch 200B amplifier (Molecular Devices) controlled by PatchMaster software (v2x91, HEKA Elektronik) via an ITC-18 A/D board (HEKA Instruments Inc). Solutions for voltage-clamp recordings: internal (in mM) 140 KCl, 13.5 NaCl, 1.8 $MgCl_2$ 0.09 EGTA, 4 MgATP, 0.3 $Na_2$GTP, 9 HEPES pH adjusted to 7.2 with KOH, 326 mOsm. The external solution contained (in mM) 3.5 KCl, 155 NaCl, 1 $MgCl_2$, 1.5 $CaCl_2$, 0.01 $CdCl_2$, 10 HEPES,

10 glucose pH adjusted to 7.4 with NaOH, 325 mOsm. The calculated liquid junction potential of –4 mV between these solutions was accounted for. For voltage-clamp recordings, neurons on cover slips were placed in the same recording chamber used for CHO cell recordings and were rinsed with an external patching solution. Neurons from *Mrgprd*[GFP] mice with green fluorescence were selected for recordings. Human DRG neurons with cell capacitances between 22.5 and 60 pF were used. After whole-cell voltage clamp was established, non-Kv2/KvS conductances were suppressed by changing to an external solution containing a cocktail of inhibitors: 100 nM alpha-dendrotoxin (Alomone) to block Kv1 (*Harvey and Robertson, 2004*), 3 μM AmmTX3 (Alomone) to block Kv4 (*Maffie et al., 2013*; *Pathak et al., 2016*), 100 μM 4-aminopyridine to block Kv3 (*Coetzee et al., 1999*; *Gutman et al., 2005*), 1 μM TTX to block TTX sensitive Nav channels, and 10 μM A-803467 (Tocris) to block Nav1.8 (*Jarvis et al., 2007*). It is possible that off target effects of blockers may introduce errors in the quantification Kv2/KvS heteromer-mediated K[+] currents. For example, 4-aminopyridine is expected to block a small fraction, 2%, of Kv2 homomers and have a lesser impact on Kv2/KvS heteromers (*Post et al., 1996*; *Thorneloe and Nelson, 2003*; *Stas et al., 2015*) which could result in a slight overestimation of the ratio of Kv2/KvS heteromers to Kv2 homomers. After addition of 1 μM RY785, neurons were given 10 steps to –24 mV for 500ms to allow for voltage dependent block of RY785. Thin-wall borosilicate glass recording pipettes were pulled with blunt tips, coated with silicone elastomer, heat cured, and tip fire-polished to resistances less than 2 MΩ. Series resistance of 1–4 MΩ was estimated from the whole-cell parameters circuit. Series resistance compensation between 55 and 98% was used to constrain voltage error to less than 15 mV. Ohmic leak was not subtracted. Neurons were held at a membrane potential of –74 mV to mimic a physiological resting potential. KvS subunits can profoundly shift the voltage-inactivation relation (*Salinas et al., 1997a*; *Kramer et al., 1998*; *Kerschensteiner and Stocker, 1999*) and this potential is likely insufficiently negative to relieve inactivation from all Kv2/KvS heteromeric channels. Also, the activation membrane potential is close to the half-maximal point of Kv2/KvS conductances. Thus the ratio of Kv2-like to KvS-like conductance is expected to vary with voltage protocols.

Whole cell recordings from *Calca*[GFP] mice were made using the same methods as *Mrgprd*[GFP] mice with the following exceptions. The internal solution contained in mM: 139.5 KGluconate, 1.6 MgCl$_2$, 1 EGTA, 0.09 CaCl$_2$, 9 HEPES, 14 creatine phosphate, 4 MgATP, 0.3 GTP pH adjusted to 7.2 with KOH. The external solution contained in mM: 155 NaCl, 3.5 KCl, 1 MgCl$_2$, 1.5 CaCl$_2$, 10 glucose, 10 HEPES pH adjusted to 7.4 with NaOH and did not contain the cocktail of inhibitors used for recordings with *Mrgprd*[GFP] mice. The liquid junction potential of –13 mV between these solutions was accounted for. Ohmic leak was subtracted. After establishing whole-cell recording, cells were lifted and placed in front of a series of quartz fiber flow pipes for rapid solution exchange and application of RY785 and GxTX.

Whole cell recordings from mouse superior cervical ganglion neurons were performed using an Axon Instruments Multiclamp 700B Amplifier (Molecular Devices). Electrodes were pulled on a Sutter P-97 puller (Sutter Instruments) and shanks were wrapped with Parafilm (American National Can Company) to allow optimal series resistance compensation without oscillation. Voltage or current commands were delivered and signals were recorded using a Digidata 1321 A data acquisition system (Molecular Devices) controlled by pCLAMP 9.2 software (Molecular Devices). The internal solution was (in mM): 140 mM K aspartate, 13.5 mM NaCl, 1.8 mM MgCl$_2$, 0.09 mM EGTA, 9 mM HEPES, 14 mM creatine phosphate (Tris salt), 4 mM MgATP, 0.3 mM Tris-GTP, pH 7.2 adjusted with KOH. The base external solution was the same as for DRG recordings. The calculated liquid junction potential of –15 mV between these solutions was accounted for. After establishing whole-cell recording, the cell was lifted and placed in front of a series of quartz fiber flow pipes for rapid solution exchange and application of RY785 and GxTX. The external solution used for recording Kv2 currents used the same cocktail of inhibitors for sodium channels and other potassium channels as for the DRG recordings except that A-803467 was omitted because the sodium current in SCG neurons is all TTX sensitive (*Toledo-Aral et al., 1997*).

## Voltage clamp analysis

Activation kinetics were fit from 10% to 90% of current ($I_K$) rise with the power of an exponential function:

$$I_K = A \left( 1 - e^{\frac{-t}{\tau_{act}}} \right)^{\sigma} \qquad (1)$$

where $A$ is the maximum current amplitude, $\tau_{act}$ is the time constant of activation, σ is sigmoidicity, and $t$ is time. The $t = 0$ mark was adjusted to 100 μs after the start of the voltage step from the holding potential to correct for filter delay and cell charging.

Conductance values were determined from tail current levels at –9 mV after 200ms steps to the indicated voltage. Tail currents were the mean current amplitude from 1 to 5ms into the –9 mV step. Conductance–voltage relations were fit with the Boltzmann function:

$$f(V) = A \left( 1 + e^{-\left(V - V_{\frac{1}{2}}\right)\frac{zF}{RT}} \right)^{-1} \qquad (2)$$

where $V$ is voltage, $A$ is amplitude, $z$ is the number of elementary charges, $F$ is Faraday's constant, $R$ is the universal gas constant, and $T$ is temperature (held at 295 K).

Deactivation kinetics were fit with a double exponential:

$$f(x) = y_0 + A_1 e^{\frac{-(t - t_0)}{\tau_1}} + A_2 e^{\frac{-(t - t_0)}{\tau_2}} \qquad (3)$$

Where $t$ is time, $y_0$ is the initial current amplitude, $t_0$ is the start time of the exponential decay, $\tau_1$ and $\tau_2$ are the time constants, and $A_1$ and $A_2$ are the amplitudes of each component.

In CHO cells peak currents were analyzed because outward currents seem to offer the best signal/noise. In neurons, voltage gated currents remained in the toxin cocktail +RY785 and GxTX, that were sometimes unstable. To minimize complications from these currents, we restricted analysis of RY785 and GxTX subtraction experiments to tail currents at elapsed times to minimize complications from non-Kv2 endogenous voltage-gated channels which deactivate more quickly. We note that the analysis of conductance activation by using tail currents is only accurate when dealing with non-inactivating conductances. We expect that inactivation of Kv2/KvS conductances during the 200ms pre-pulse is minimal (*Salinas et al., 1997a*; *Kramer et al., 1998*; *Kerschensteiner and Stocker, 1999*) and did not notice inactivation during the activation pulse. Also, deactivation kinetics can vary in a heterogenous population of Kv2/KvS heteromers. While analysis of tail currents could skew the quantification of total Kv2 like and KvS-like conductances, our data supports that mouse nociceptors and human neurons have tail currents that are resistant to RY785 and sensitive to GxTX consistent with the presence of Kv2/KvS heteromers.

## Preparation of rKv2.1 and hKv2.1/hKv8.1 docking inputs

The rKv2.1 homomer (PDB ID: 8SD3; *Fernández-Mariño et al., 2023*) was energy minimized using cryo-EM structure refinement with Rosetta (*Wang et al., 2016*). This energy-minimized model was used for generation of the hKv2.1/hKv8.1 heteromer using ColabFold (*Mirdita et al., 2022*) with 48 recycles, dropout enabled, and Amber minimization (*Salomon-Ferrer et al., 2013*). To provide the same pore conformation as input between homomer and heteromer docking, the Kv8.1 subunit from ColabFold was superimposed to a Kv2.1 subunit from the energy-minimized model used for homomer docking; this heteromer model was then energy minimized by fixing the Kv2.1 backbone and sidechain chi torsion angles while allowing the Kv8.1 backbone and sidechain chi torsion angles to minimize with Rosetta FastRelax (*Tyka et al., 2011*) using the BetaNov16 scoring function (*Pavlovicz et al., 2020*).

## Preparation of RY785 docking input

RY785 was extracted as a structure data file from PubChem (*Kim et al., 2023*). With Avogadro (*Hanwell et al., 2012*), RY785 underwent bond correction, protonation at pH 7.4, and energy minimization using the Merck molecular force field (*Halgren, 1996a*; *Halgren, 1996b*; *Halgren, 1996c*;

*Halgren, 1996d*; *Halgren and Nachbar, 1996e*; *Halgren, 1999a*; *Halgren, 1999b*). Next, using the Antechamber protocol of AmberTools (*Case et al., 2023*), the partial atomic charge, atom, and bond-type assignments for each ligand were AM1-BCC corrected. The input conformer was generated using the RosettaGenFF (*Park et al., 2021*) crystal structure prediction protocol, taking the lowest energy packing arrangement as input.

### RY785 docking
Docking was performed using Rosetta's GALigandDock (*Park et al., 2021*) for ion channel docking as described previously (*Harris et al., 2024*). Briefly, GALigandDock in the flexible docking mode was used with a padding value of 7 Å, 20 generations with a pool of 100 poses, and the entire pool of poses as output. With this run mode, the entire pore cavity had the potential to be sampled with side chain flexibility. We ran GALigandDock 100 times, generating 10,000 total models for both the rKv2.1 homomer and hKv2.1/hKv8.1 heteromer. The top 10 models were selected for visual inspection by sorting the top 10% of models by total score, followed with the top 10 of the subset by interface score. Contact analysis was performed using UCSF ChimeraX (*Pettersen et al., 2021*) Clashes and Contacts tool and the H-Bonds tool with default distance cutoff and acceptance settings.

### Immunofluorescence
Kv2.1-CHO cells were fixed for 15 min at 4 °C in 4% formaldehyde prepared fresh from paraformaldehyde in PBS buffer pH 7.4. Cells were then washed 3x5 min in PBS, followed by blocking in blotto-PBS (PBS, pH 7.4 with 4% (w/v) non-fat milk powder and 0.1% (v/v) Triton-X100) for 1 hr. Cells were incubated for 1 hr with primary antibodies diluted in blotto-PBS and subsequently washed 3x5 min in PBS. Antibodies used were mAb K89/34 for Kv2.1 (NeuroMab, RRID:AB_1067225), rabbit pAb 5.1 C for Kv5.1 (in-house, RRID:AB_3076240), and rabbit anti-V5 for Kv9.3-V5 (Rockland, 600-401-378). For surface labeling of Kv5.1, non-permeabilized cells were incubated with Kv5.1 mAb (Santa Cruz Biotech, 81881) in blotto-PBS lacking Triton-X100. The cells were then incubated with mouse IgG subclass- and/or species-specific Alexa-conjugated fluorescent secondary antibodies (Invitrogen) diluted in blotto-PBS for 45 min and washed 3x5 min in PBS. Cover glasses were mounted on microscope slides with Prolong Gold mounting medium (Thermo Fisher, P36930) according to the manufacturer's instructions. Widefield fluorescence images were acquired with an AxioCam MRm digital camera installed on a Zeiss AxioImager M2 microscope with a 63×/1.40 NA Plan-Apochromat oil immersion objective and an ApoTome coupled to Axiovision software version 4.8.2.0 (Zeiss, Oberkochen, Germany).

### Multiplex in situ hybridization
A 6-week-old *Mrgprd^GFP* mouse was briefly anesthetized with 3–5% isoflurane and then decapitated. The spinal column was dissected, and the left and right L1 DRG were removed and drop fixed for 12 min in ice cold 4% paraformaldehyde in 0.1 M phosphate buffer (PB) pH adjusted to 7.4. The L1 vertebrae was identified by the 13th rib. The DRG was washed 3×10 min each in PB and cryoprotected at 4 °C in 30% sucrose diluted in PB for 2 hr. The DRG were then frozen in Optimal Cutting Temperature (OCT) compound (Fisher, 4585) and stored at –80 °C until sectioning. Samples were cut into 20 µm sections on a freezing stage sliding microtome and were collected on Colorfrost Plus microscope slides (Thermo Fisher Scientific, 12-550-19). Sections were processed for RNA in situ detection using an RNAscope Fluorescent Detection Kit according to the manufacturer's instructions (Advanced Cell Diagnostics) with the following probes: *KCNF1* (508731, mouse) or *KCNS1* (525941, mouse). TSA Vivid 650 Fluorophore was used to label probes (TSA Vivid, 7527). Following in situ hybridization, immunohistochemistry to label GFP was performed. Sections were incubated in vehicle solution (4% milk, 0.2% triton diluted in PB) for 1 hr at RT. Tissue was then incubated in a rabbit polyclonal anti-GFP antibody (Rockland 600-401-215S) diluted 1:1000 in vehicle overnight at 4 °C. Sections were washed three times in vehicle for 5 min per wash and then incubated in a goat anti-rabbit secondary antibody (Invitrogen, A-11008) diluted 1:1500 in vehicle. Sections were then mounted with Prolong Gold (Thermo Fisher, P36930) and #1.5 cover glass (Thermo Fisher Scientific, NC1776158).

## Imaging

Images were acquired with an inverted scanning confocal and airy disk imaging system (Zeiss LSM 880 Airyscan, 410900-247-075) run by ZEN black v2.1. Laser lines were 488 nm and 633 nm. Images were acquired with a 0.8 NA 20 x objective (Zeiss, 420650–9901) details in figure legends.

## Statistics

All statistical tests were performed in Igor Pro software version 8 (Wavemetrics, Lake Oswego, OR). Independent replicates (n) are individual cells/neurons while biological replicates (N) are individual mice. All tests were two-tailed. Wilcoxon rank tests were used for two-sample comparisons. Dunnett tests were used for multiple comparisons.

## Acknowledgements

We thank the human tissue donors and their families for their generous donations. We thank Sierra Donor Services for recovering human dorsal root ganglia, as well as Sean Van Slyck, Marnae Salampessy, and Theanne Griffith for helping arrange for human tissue. We thank Bryan Copits, Ted Price, and Juliet Mwirigi for advice on culturing human neurons. We thank Cyrrus Espino, Hai Nguyen, and Geir Hareland for preparation of human tissues. We thank Josh Tulman for illustrations. We thank Bruce Bean for scientific discussions and feedback on the manuscript. GxTx-Nle35 was synthesized at the Molecular Foundry of the Lawrence Berkeley National Laboratory under U.S. Department of Energy contract DE-AC02-05CH11231. Research at the University of California Davis was supported by the University of California Davis and U.S. National Institutes of Health grant R03-TR004200. Research at Harvard was supported by National Institutes of Health grant R35-NS127216. The Neurobiology Course at the Marine Biological Laboratory is supported by National Institutes of Health grant R25-NS063307.

## Additional information

### Competing interests

Jon T Sack: Reviewing editor, eLife. The other authors declare that no competing interests exist.

### Funding

| Funder | Grant reference number | Author |
|---|---|---|
| University of California Davis | R03-TR004200 | Robert G Stewart<br>Matthew James Marquis<br>Michael Ferns<br>Jon T Sack |
| National Institutes of Health | R35-NS127216 | Robert G Stewart<br>Sooyeon Jo |
| National Institutes of Health | R25-NS063307 | Robert G Stewart<br>Aman S Aberra<br>Verity Cook<br>Zachary Whiddon<br>Jon T Sack |
| National Institutes of Health | R03-TR004200 | Robert G Stewart<br>Matthew James Marquis<br>Michael Ferns<br>Jon T Sack |

The funders had no role in study design, data collection and interpretation, or the decision to submit the work for publicatio

### Author contributions

Robert G Stewart, Matthew James Marquis, Brandon J Harris, Conceptualization, Data curation, Formal analysis, Investigation, Visualization, Methodology, Writing – original draft, Writing – review and editing; Sooyeon Jo, Data curation, Investigation, Writing – review and editing; Aman S Aberra,

Verity Cook, Zachary Whiddon, Conceptualization, Data curation, Investigation, Writing – review and editing; Vladimir Yarov-Yarovoy, Conceptualization, Funding acquisition, Investigation, Writing – review and editing; Michael Ferns, Conceptualization, Data curation, Formal analysis, Funding acquisition, Investigation, Writing – review and editing; Jon T Sack, Conceptualization, Data curation, Formal analysis, Supervision, Funding acquisition, Investigation, Visualization, Methodology, Writing – original draft, Project administration, Writing – review and editing

#### Author ORCIDs
Robert G Stewart ![orcid] https://orcid.org/0009-0003-5407-0346
Matthew James Marquis ![orcid] https://orcid.org/0000-0002-5556-9072
Brandon J Harris ![orcid] https://orcid.org/0000-0003-3894-0180
Aman S Aberra ![orcid] https://orcid.org/0000-0002-4805-541X
Vladimir Yarov-Yarovoy ![orcid] https://orcid.org/0000-0002-2325-4834
Michael Ferns ![orcid] https://orcid.org/0000-0002-1545-3024
Jon T Sack ![orcid] https://orcid.org/0000-0002-6975-982X

#### Ethics
Studies were approved by the Institutional Animal Care and Use Committees (IACUC) at UC Davis (protocol 23621) and Harvard Medical School (protocol IS00001369-6) and conform to guidelines established by the NIH.

Reviewer #1 (Public review): https://doi.org/10.7554/eLife.99410.3.sa1
Reviewer #2 (Public review): https://doi.org/10.7554/eLife.99410.3.sa2
Author response https://doi.org/10.7554/eLife.99410.3.sa3

## Additional files

#### Supplementary files
Source data 1. Source data contains IgorPro files with data presented in this manuscript.

#### Data availability
All data analyzed for this study are included in the manuscript. IgorPro (.pxp) source data files provided are part of the Kv2.1/Kv5.1 dataset was presented previously (*Ferns et al., 2025*).

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
