## [Editor Report · eLife Assessment]

Some delayed rectifier currents in neurons are formed by the combination of Kv2 and silent subunits, KvS. However, we lack the tools to identify these heteromeric channels in vivo. In this **important** study by the Sack group, the authors identify a pharmacological tool that can reveal the presence of KvS subunits as components of the delayed rectifier potassium currents in selected neurons. The experimental evidence presented in the manuscript is **compelling** and represents a significant advance that should be of interest to a wide community of neuroscientists and channel physiologists.

---

## [Referee Report · Reviewer #1 (Public review)]

Summary:

Kv2 subfamily potassium channels contribute to delayed rectifier currents in virtually all mammalian neurons and are encoded by two distinct types of subunits: Kv2 alpha subunits that have the capacity to form homomeric channels (Kv2.1 and Kv2.2), and KvS or silent subunits (Kv5,6,8.9) that can assemble with Kv2.1 or Kv2.2 to form heteromeric channels with novel biophysical properties. Many neurons express both types of subunits and therefore have the capacity to make both homomeric Kv2 channels and heteromeric Kv2/KvS channels. Determining the contributions of each of these channel types to native potassium currents has been very difficult because the differences in biophysical properties are modest and there are no Kv2/KvS-specific pharmacological tools. The authors set out to design a strategy to separate Kv2 and Kv2/KvS currents in native neurons based on their observation that Kv2/KvS channels have little sensitivity to the Kv2 pore blocker RY785 but are blocked by the Kv2 VSD blocker GxTx. They clearly demonstrate that Kv2/KvS currents can be differentiated from Kv2 currents in native neurons using a two-step strategy to first selectively block Kv2 with RY785, and then block both with GxTx. The manuscript is beautifully written; takes a very complex problem and strategy and breaks it down so both channel experts and the broad neuroscience community can understand it.

Strengths:

The compounds the authors use are highly selective and unlikely to have significant confounding cross-reactivity to other channel types. The authors provide strong evidence that all Kv2/KvS channels are resistant to RY785. This is a strength of the strategy - it can likely identify Kv2/KvS channels containing any of the 10 mammalian KvS subunits and thus be used as a general reagent on all types of neurons. The limitation then of course is that it can't differentiate the subtypes, but at this stage, the field really just needs to know how much Kv2/KvS channels contribute to native currents and this strategy provides a sound way to do so.

Weaknesses:

The authors are very clear about the limitations of their strategy, the most important of which is that they can't differentiate different subunit combinations of Kv2/KvS heteromers. This study is meant to be a start to understanding the roles of Kv2/KvS channels in vivo. As such, this is a minor weakness, far outweighed by the potential of the strategy to move the field through a roadblock that has existed since its inception.

The study accomplishes exactly what it set out to do: provide a means to determine the relative contributions of homomeric Kv2 and heteromeric Kv2/KvS channels to native delayed rectifier K+ currents in neurons. It also does a fabulous job laying out the case for why this is important to do.

Comments on revisions:

I liked this manuscript the first time and thought it was a great attempt to address a difficult problem, made more difficult by confusing background literature and conventions. The authors have kept all the strong points I liked from the first round and made it even stronger with their thoughtful and substantive responses to reviews. My first review was strongly supportive, and my initial short assessment/public review was written with the assumption that they would be public and the paper would be published essentially in its original form. All those points still apply so I am going to leave the initial reviews as is. The paper is a pleasure to read and a nice contribution to the field.

---

## [Referee Report · Reviewer #2 (Public review)]

The authors used combined blockers/modulators to dissect the potassium currents mediated by inter-subunit heteromeric Kv channels. The method is robust given that the researchers know their limitations. Nevertheless, the authors elegantly tested their hypotheses, making this manuscript friendly to read despite the depth of all aspects they dealt with.

The quality of the data presented will positively impact the science involved in the study heteromeric channels, with clear developments in the field. Finally, the approach presented may unlock new studies related to these channels.

Comments on revisions:

The authors clarified all my points and beyond, specifically by adding some computational work that will also contribute to the subfield of heteromeric Kv channels.

---

## [Author Response]

The following is the authors’ response to the original reviews

**Public Reviews:**

**Reviewer #1 (Public Review):**
Summary:Kv2 subfamily potassium channels contribute to delayed rectifier currents in virtually all mammalian neurons and are encoded by two distinct types of subunits: Kv2 alpha subunits that have the capacity to form homomeric channels (Kv2.1 and Kv2.2), and KvS or silent subunits (Kv5,6,8.9) that can assemble with Kv2.1 or Kv2.2 to form heteromeric channels with novel biophysical properties. Many neurons express both types of subunits and therefore have the capacity to make both homomeric Kv2 channels and heteromeric Kv2/KvS channels. Determining the contributions of each of these channel types to native potassium currents has been very difficult because the differences in biophysical properties are modest and there are no Kv2/KvS-specific pharmacological tools. The authors set out to design a strategy to separate Kv2 and Kv2/KvS currents in native neurons based on their observation that Kv2/KvS channels have little sensitivity to the Kv2 pore blocker RY785 but are blocked by the Kv2 VSD blocker GxTx. They clearly demonstrate that Kv2/KvS currents can be differentiated from Kv2 currents in native neurons using a two-step strategy to first selectively block Kv2 with RY785, and then block both with GxTx. The manuscript is beautifully written; takes a very complex problem and strategy and breaks it down so both channel experts and the broad neuroscience community can understand it.Strengths:The compounds the authors use are highly selective and unlikely to have significant confounding cross-reactivity to other channel types. The authors provide strong evidence that all Kv2/KvS channels are resistant to RY785. This is a strength of the strategy - it can likely identify Kv2/KvS channels containing any of the 10 mammalian KvS subunits and thus be used as a general reagent on all types of neurons. The limitation then of course is that it can't differentiate the subtypes, but at this stage, the field really just needs to know how much Kv2/KvS channels contribute to native currents and this strategy provides a sound way to do so.Weaknesses:The authors are very clear about the limitations of their strategy, the most important of which is that they can't differentiate different subunit combinations of Kv2/KvS heteromers. This study is meant to be a start to understanding the roles of Kv2/KvS channels in vivo. As such, this is a minor weakness, far outweighed by the potential of the strategy to move the field through a roadblock that has existed since its inception.The study accomplishes exactly what it set out to do: provide a means to determine the relative contributions of homomeric Kv2 and heteromeric Kv2/KvS channels to native delayed rectifier K+ currents in neurons. It also does a fabulous job laying out the case for why this is important to do.
**Reviewer #2 (Public Review):**
Summary:Silent Kv subunits and the channels containing these Kv subunits (Kv2/KvS heteromers) are in the process of discovery. It is believed that these channels fine-tune the voltage-activated K+ currents that repolarize the membrane potential during action potentials, with a direct effect on cell excitability, mostly by determining action potentials firing frequency.Strengths:What makes silent Kv subunits even more important is that, by being expressed in specific tissues and cell types, different silent Kv subunits may have the ability to fine-tune the delayed rectifying voltage-activated K+ currents that are one of the currents that crucially determine cell excitability in these cells. The present manuscript introduces a pharmacological method to dissect the voltage-activated K+ currents mediated by Kv2/KvS heteromers as a means of starting to unveil their importance, together with Kv2-only channels, to the cells where they are expressed.Weaknesses:While the method is effective in quantifying these currents in any isolated cell under an electric voltage clamp, it is ineffective as a modulating maneuver to perhaps address these currents in an in vivo experimental setting. This is an important point but is not a claim made by the authors.

We agree. We have now stated in the introduction that this study does not address the roles of Kv2/KvS currents in an in vivo setting.

Manuscript revisions:

While this study does not address the impact of GxTX or RY785 on action potentials or in vivo, the distinct pharmacology of Kv2/KvS heteromers presented here suggests that KvS conductances could be targeted to selectively modulate discrete subsets of cell types.

There are other caveats with the methods and data:(i) The need for a 'cocktail' of blockers to supposedly isolate Kv2 homomers and Kv2/KvS heteromers' currents from others may introduce errors in the quantification Kv2/KvS heteromers-mediated K+ currents and that is due to possible blockers off targets.

We now point out that is possible that off target effects of blockers may introduce errors, include references that identify the selectivity of the blockers used in the cocktail, and specifically note that 4-aminopyridine in the cocktail is expected to block 2% of Kv2 homomers yet have a lesser impact Kv2/KvS heteromers. Additionally, to test whether the KvS isolation strategy requires the cocktail in neurons, we performed new experiments on a different subclass of nociceptors without the blocker cocktail and identified a substantial KvS-like component (new Fig 7 Supplement 3).

Manuscript revisions:

“After whole-cell voltage clamp was established, non-Kv2/KvS conductances were suppressed by changing to an external solution containing a cocktail of inhibitors: 100 nM alpha-dendrotoxin (Alomone) to block Kv1 (Harvey and Robertson, 2004), 3 μM AmmTX3 (Alomone) to block Kv4 (Maffie et al., 2013; Pathak et al., 2016), 100 μM 4-aminopyridine to block Kv3 (Coetzee et al., 1999; Gutman et al., 2005), 1 μM TTX to block TTX sensitive Nav channels, and 10 μM A803467 (Tocris) to block Nav1.8 (Jarvis et al., 2007). It is possible that off target effects of blockers may introduce errors in the quantification Kv2/KvS heteromer-mediated K^+^ currents. For example, 4-aminopyridine is expected to block a small fraction, 2%, of Kv2 homomers and have a lesser impact on Kv2/KvS heteromers (Post et al., 1996; Thorneloe and Nelson, 2003; Stas et al., 2015) which could result in a slight overestimation of the ratio of Kv2/KvS heteromers to Kv2 homomers.”

“We also tested the other major mouse C-fiber nociceptor population, peptidergic nociceptors, to determine if this subpopulation also has conductances resistant to RY785 yet sensitive to GxTX. We voltage clamped DRG neurons from a *CGRPGFP* mouse line that expresses GFP in peptidergic nociceptors (Gong et al., 2003). Deep sequencing has identified mRNA transcripts for Kv6.2, Kv6.3, Kv8.1 and Kv9.3 present in GFP+ neurons, an overlapping but distinct set of KvS subunits from the *MrgprdGFP* non-peptidergic population (Zheng et al., 2019). In GFP+ neurons from *CGRPGFP* mice, we found that a fraction of outward current was inhibited by 1 µM RY785 and additional current inhibited by 100 nM GxTX (Fig 7 Supplement 3 A-C). In these experiments, 58 ± 2% (mean ± SEM) was KvS-like (Fig 7 Supplement 3 D) identifying that KvSlike conductances are present in these peptidergic nociceptors. For *CGRPGFP* neurons we did not include the Kv1, Kv3, Kv4, Nav and Cav channel inhibitor cocktail used for other neuron experiments, indicating that the cocktail of inhibitors is not required to identify KvS-like conductances.”

(ii) During the electrophysiology experiments, the authors use a holding potential that is not as negative as it is needed for the recording of the full population of the Kv2/KvS channels. Depolarized holding potentials lead to a certain level of inactivation of the channels, that vary according to the KvS involved/present in that specific population of channels. As a reminder, some KvS promote inactivation and others prevent inactivation. Therefore, the data must be interpreted as such.

We agree. We now point out that the physiological holding potentials used are insufficiently negative to relieve inactivation from all Kv2/KvS heteromeric channels. We also note that the ratio of Kv2-like to KvS-like conductance is expected to vary with voltage protocols.

Manuscript revisions:

“Neurons were held at a membrane potential of –74 mV to mimic a physiological resting potential. KvS subunits can profoundly shift the voltage-inactivation relation (Salinas et al., 1997a; Kramer et al., 1998; Kerschensteiner and Stocker, 1999) and this potential is likely insufficiently negative to relieve inactivation from all Kv2/KvS heteromeric channels. Also, the activation membrane potential is close to the half-maximal point of Kv2/KvS conductances. Thus the ratio of Kv2-like to KvS-like conductance is expected to vary with voltage protocols.”

(iii) The analysis of conductance activation by using tail currents is only accurate when dealing with non-inactivating conductances. Also, in dealing with a heterogenous population of Kv2/KvS heteromers, heterogenous K+ conductance deactivation kinetics is a must. Indeed, different KvS may significantly relate to different deactivation kinetics as well.

We now discuss that the bi-exponential fit of tail currents is likely inadequate to capture the deactivation kinetics of all underlying components of a heterogenous population of Kv2/KvS heteromers.

Manuscript revisions:

“We note that the analysis of conductance activation by using tail currents is only accurate when dealing with non-inactivating conductances. We expect that inactivation of Kv2/KvS conductances during the 200 ms pre-pulse is minimal (Salinas et al., 1997a; Kramer et al., 1998; Kerschensteiner and Stocker, 1999) and did not notice inactivation during the activation pulse. Also, deactivation kinetics can vary in a heterogenous population of Kv2/KvS heteromers. While analysis of tail currents could skew the quantification of total Kv2 like and KvS-like conductances, our data supports that mouse nociceptors and human neurons have tail currents that are resistant to RY785 and sensitive to GxTX consistent with the presence of Kv2/KvS heteromers.”

(iv) Silent Kv subunits may be retained in the ER, in heterologous systems like CHO cells. This aspect may subestimate their expression in these systems. Nevertheless, the authors show similar data in CHO cells and in primary neurons.

We agree. We now note that in heterologous systems, including CHO cells, transfection of KvS subunits can result in KvS subunits that are retained intracellularly.

Manuscript revisions:

“While a fraction of KvS subunits appear to be retained intracellularly, immunofluorescence for Kv5.1, Kv9.3 and Kv2.1 also appeared localized to the perimeter of transfected Kv2.1-CHO cells (Figure 1 Supplement).”

(v) The hallmark of silent Kv subunits is their effect on the time inactivation of K+ currents. As such, data should be shown throughout, preferably, from this perspective, but it was only done so in Figure 4G.

Indeed, effects on inactivation are a hallmark of KvS subunits. However, quantifying inactivation of Kv2/KvS channels requires steps to positive voltages for approximately 10 seconds. In neurons steps this long usually resulted in irreversible changes in leak currents/input resistance that degraded the accuracy of RY785/GxTX subtraction currents. Consequently, we did not acquire inactivation data in neurons, and we now explain in the manuscript why such data was not obtained.

Manuscript revisions:

“While changes in inactivation are prominent with KvS subunits, we did not investigate inactivation in neurons because the lengthy depolarizations required often resulted in irreversible leak current increases that degraded the accuracy of RY785/GxTX subtraction current quantification.”

(vi) Functional characterization of currents only, as suggested by the authors as a bona fide of Kv2 and Kv2/KvS currents, should not be solely trusted to classify the currents and their channel mediators.

We agree, and now state explicitly that functional characterization cannot be trusted to classify their channel mediators of conductances, and we try to be clear about this throughout the manuscript by using soft terms such as "KvS-like" when identity is uncertain.

Manuscript revisions:

“As functional characterization alone cannot be trusted to classify their channel mediators of conductances, we define conductances consistent with Kv2/KvS heteromers as 'KvS-like' and conductances consistent with Kv2 homomers as 'Kv2-like'.”

**Recommendations for the authors:**

**Reviewer #1 (Recommendations For The Authors):**
There is not a lot to do here - this was a real pleasure to read and very easy to understand, as written. Here are a few minor things to consider:(1) The naming of the KvS subunits has always been confusing - it is not clear that Kv5,6,8,9 are members of the Kv2 subfamily from the names. KvS does a good job of differentiating them by assembly phenotype and has been used a lot in the literature, but it doesn't solve the misconception of what subfamily they belong to. This might not matter so much for mammals, where all KvS channels are in the Kv2 subfamily, but it makes it impossible to extend the naming system to other animals where subunits requiring heteromeric assembly are common in most subfamilies. How about trying the name Kv2S? It would have continuity with KvS in the reader's mind, make it clear that they are Kv2 subfamily, and make a naming system that could be extended beyond vertebrates. This is not a problem the authors created - just a completely optional suggestion on how to solve it if so inclined.

We agree that naming conventions for these subunits are problematic, and agonized quite a bit about nomenclature. In the end we chose to stick with the precedent of KvS.

(2) Another naming issue they should definitely change is the use of "subfamily" for the different KvS subtypes (Kv5, Kv6, Kv8, and Kv9). This really creates confusion with the higher-order subfamilies that have a very clear functional definition: a subfamily of Kv genes is a group of related genes that have assembly compatibility. Those are Kv1, Kv2, Kv3 and Kv4. KvS genes are assembly compatible with Kv2, evolutionarily derived from the Kv2 lineage, and thus clearly a part of the Kv2 subfamily. Using a subfamily for the next lower level of the naming hierarchy confuses this. The authors should use different terms like sub-type or class or subgroups for the divisions within KvS.

Thank you. We have standardized to Kv2/KvS as a subfamily; Kv5, Kv6, Kv8, and Kv9 as subtypes; and individual proteins, e.g. Kv8.1, as subunits.

(3) When you discuss whether the KvS subunit directly disrupts Ry785 binding in the pore or works allosterically and you said you know which KvS residues point into the pore from models, I thought that maybe you could tell from a sequence alignment whether the KvS channels you didn't test look the same in the conduction pathway as the ones you did test. If so, you could mention that if the binding site is the pore, they should all be resistant. Alternatively, if one you didn't test looks fundamentally more similar to the Kv2s in this region, then maybe it could be fingered as a possible exception that needs to be tested later.

Great ideas. We now assess sequence KvS variability near the proposed RY785 binding site in all KvS subunits. We generated structural models of RY785 docking to Kv2.1 and Kv2.1/Kv8.1 and found that residues near RY785 are different in all KvS subunits.

Manuscript revisions:

“We analyzed computational structural models of RY785 docked to a Kv2.1 homomer and a 3:1 Kv2.1:Kv8.1 heteromer (Fig 9) to gain structural insight into how KvS subunits might interfere with RY785 binding. We used Rosetta to dock RY785 to a cryo-EM structure of a Kv2.1 homomer in an apparently open state (Fernández-Mariño et al., 2023). The top-scoring docking pose has RY785 positioned below the selectivity filter and off-axis of the pore (Fig 9 A), similar to a stable pose observed in molecular dynamic simulations (Zhang et al., 2024). In this pose, RY785 contacts a collection of Kv2.1 residues that vary in every KvS subtype (Fig 9 B,D,E). Notably, RY785 bound similarly to a 3:1 model of Kv2.1/Kv8.1, in contact with the three Kv2.1 subunits, yet avoided the Kv8.1 subunit (Fig 9C). This is consistent with RY785 binding less well to Kv2.1/Kv8.1 heteromers, and also suggests that a 3:1 Kv2:KvS channel could retain a RY785 binding site when open.”

(4) Future suggestion or tip - not for this paper. Your data shows your isolation strategy works really well on Kv6 channels, and these are also the Kv2/KvS channels that have the most pronounced biophysical changes. Working on neurons that have a prominent Kv2/Kv6 component would really show how well the strategy outlined here works to describe the physiology of native neurons. The highest KvS expression I have seen in public data in a wellstudied cell type is Kv6.4 in spinal motor neurons.

Wonderful tip, thank you. We are indeed very interested in Kv6.4 in spinal motor neurons.

**Reviewer #2 (Recommendations For The Authors):**
The manuscript makes a good contribution to the identification of Kv2/KvS channels in primary cells. The pharmacological method proposed by the authors to dissect the currents in an experimental setting seems proper. Although meritorious in themselves, the findings are heavily phenomenological in the opinion of this reviewer. The manuscript should be improved with some level of mechanistic data and/or the demonstration of different levels of expression in different cell types.

Thank you for the suggestions. This manuscript now demonstrates strikingly higher levels of the KvS-like component of Kv2 currents in somatosensory (DRG nonpeptidergic and peptidergic nociceptor) versus autonomic (SCG) neuron types. The mechanistic question of what electrophysiological properties the KvS subunits are providing to the neuronal circuit is an exciting one that we are pursuing separately.

Manuscript revisions:

“While we found only RY785-sensitive Kv2-like conductances in SCG neurons, Kv2/KvS heteromer-like conductances were dominant in DRG neurons.”

At present, the manuscript says that the combination of RY785 and guangxitoxin-1E can be used to define Kv2/KvS-mediated K+ currents. Importantly, this method cannot be used in a way that one can functionally determine the function of Kv2/KvS channels, since it depends on the pre-blocking of Kv2-mediated K+ currents prior. In the opinion of this reviewer, this fact decreases the attention of a potential reader.

Indeed, our study is focused on revealing KvS heteromers by voltage clamp, and we now clarify in the introduction that we do not determine the function of Kv2/KvS channels in this study, so as not to lead the reader to expect studies of neuronal signaling.

However, the selective pharmacology we identify suggests RY785 application could reveal the function of Kv2 homomers, and for RY785-insensitive signaling, GxTX application of could reveal the function of Kv2/KvS heteromers. We now mention these possible applications in the Discussion.

Manuscript revisions:

“While this study does not address the impact of GxTX or RY785 on action potentials or in vivo, the distinct pharmacology of Kv2/KvS heteromers presented here suggests that KvS conductances could be targeted to selectively modulate discrete subsets of cell types.”

Please find below suggestions for improving the manuscript:(1) The term "Kv2/KvS heteromers" should be used throughout instead of variations such as "Kv2/KvS channels", "Kv2/KvS" and others. Standardization of the term to refer to heteromers would make the manuscript easier to read.

Thank you. We have standardized terms to consistently refer to Kv2/KvS heteromers.

(2) Confusing terms like KvS conductances, KvS-like conductances, KvS-like (RY785-resistant, GxTX-sensitive) currents, and KvS channels should be avoided because they disregard the current belief that KvS cannot form functional homomeric channels. The term KvS-containing channels, and Kv2/KvS channels, seem more accurate. Uniformization in this regard will also make the manuscript more easily readable.

Thank you. We have standardized terms to Kv2/KvS heteromers and KvS-containing channels when channel subunits are known and the use terms KvS-like and Kv2-like for functionally identified endogenous conductances with unknown channel subunits.

(3) Referring to KvS as a regulatory subunit is inaccurate. It is clear that KvS is part of, and it makes up the alpha pore. KvS therefore is a part of the conductive pathway and not a regulatory (suggesting accessory) subunit. KvS take part in selectivity filter (fully conserved), but they also make up an important part of the conducting pathway with non-conserved amino acid residues.

We felt it important to include the descriptor “regulatory” to connect our nomenclature with prior use of the descriptor in the literature, and now only use the term at the start of the introduction.

Manuscript revisions:

“A potential source of molecular diversity for Kv2 channels are a group of Kv2-related proteins which have been referred to as regulatory, silent, or KvS subunits.”

(4) The use of a cocktail of channel inhibitors may affect the quantification of Kv2/KvS heteromers-mediated K+ currents because they may interact with RY785 and/or GxTx or they may even interact with the sites for these two drugs on Kv2-containing channels.

This is an interesting point worth considering, thank you. We now alert readers to this possibility in the discussion when considering the limitations of our approach.

Manuscript revisions:

“Also, the cocktail of inhibitors used in most neuron experiments here could potentially alter RY785 or GxTX action against KvS/Kv2 channels.”

(5) The graphical representation of fractional blocking and other parameters (e.g., Fig 1D), is hard to read in these slim plots. In my opinion, tall bars would be more meaningfully visualized.

Thank you for pointing out that the graphs were hard to read, we have made the graph easier to read and added tall bars.

(6) Vehicle control for IHC and electrophysiology. Please state what is the vehicle used in the electrophysiology experiments.

Thank you. The composition of vehicle has now been stated in the methods.

Manuscript revisions:

“All RY785 solutions contained 0.1% DMSO. Vehicle control solutions also contained 0.1% DMSO but lacked RY785.”

“Sections were incubated in vehicle solution (4% milk, 0.2% triton diluted in PB) for 1 hr at RT.”

(7) The reference Trapani & Korn, 2003 (?) is not included in the list. This reference is important since it sets what are the Kv2.1-CHO cells. In this regard it is also important to mention, even better to address, the expressing qualities of this system in the face of a co-expression with a plasmid-based expression of silent Kv subunits. Are these two ways of expressing Kv subunits, meant to come together (or not) in heteromers, balanced? This question is critical here. Still, in regard to Kv2.1-CHO cells, it was not clear in the manuscript if the term "transfection" refers only to the plasmids used to temporarily induce the expression of silent Kv subunits and potentially Kv channels accessory subunits.

We now include the Trapani & Korn, 2003 reference (thank you for pointing out this accidental omission), and better explain expression methods. The benefit of the inducible Kv2.1 expression is control of Kv conductance densities which can otherwise become so large as to be refractory to voltage clamp. The beauty of the expression system is that cells recently transfected with KvS subunits can be induced to express just enough Kv2.1 to get a substantial but not clampoverwhelming RY785-resistant Kv2/KvS conductance. We also discuss that our expression methods are distinct from past studies. We stop short of comparing the expression systems, as this is beyond the scope of what we set out to study.

Manuscript revisions: See next response

(8) Kv2.1-CHO cells transfection procedures, induction, and validation are unclear. This validation is important here.

We have clarified transfection procedures, induction, and validation in the methods section.

Manuscript revisions:

“The CHO-K1 cell line transfected with a tetracycline-inducible rat Kv2.1 construct (Kv2.1-CHO) (Trapani and Korn, 2003) was cultured as described previously (Tilley et al., 2014).”

Transfections were achieved with Lipofectamine 3000 (Life Technologies, L3000001). 1 μl Lipofectamine was diluted, mixed, and incubated in 25 μl of Opti-MEM (Gibco, 31985062).”

“Concurrently, 0.5 μg of KvS or AMIGO1 or Navβ2, 0.5 μg of pEGFP, 2 μl of P3000 reagent and 25 μl of Opti-MEM were mixed. DNA and Lipofectamine 3000 mixtures were mixed and incubated at room temperature for 15 min. This transfection cocktail was added to 1 ml of culture media in a 24 well cell culture dish containing Kv2.1-CHO cells and incubated at 37 °C in 5% CO2 for 6 h before the media was replaced. Immediately after media was replaced, Kv2.1 expression was induced in Kv2.1-CHO cells with 1 μg/ml minocycline (Enzo Life Sciences, ALX380-109-M050), prepared in 70% ethanol at 2 mg/ml. Voltage clamp recordings were performed 12-24 hours later. We note that the expression method of Kv2/KvS heteromers used here is distinct from previous studies which show that the KvS:Kv2 mRNA ratio can affect the expression of functional Kv2/KvS heteromers (Salinas et al., 1997b; Pisupati et al., 2018). We validated the functional Kv2/KvS heteromer expression using voltage clamp to establish distinct channel kinetics and the presence of RY785-resistant conductance in KvS-transfected cells and using immunohistochemistry to label apparent surface localization of KvS subunits (Figure 4, Figure 1 Supplement, Figure 1 and Figure 5).”

(9) It is important for readers to add some context to Kv2.1/Kv8.1 channels (and other Kv2/KvS heteromers) used to test the combination of RY785 and GxTx. In my opinion, this enriches the discussion.

Good idea. We have added context about each of the KvS subunits we test.

Manuscript revisions:

“To test the pharmacological response of KvS we began with Kv8.1, a subunit that creates heteromers with biophysical properties distinct from Kv2 homomers (Salinas et al., 1997a), and modulates motor neuron vulnerability to cell death (Huang et al., 2024).

Each of these KvS subunits create Kv2/KvS heteromers that have distinct biophysical properties (Kramer et al., 1998; Kerschensteiner and Stocker, 1999; Bocksteins et al., 2012). Kv5.1/Kv2.1 heteromers play an important role in controlling the excitability of mouse urinary bladder smooth muscle (Malysz and Petkov, 2020), mutations in Kv6.4 have been shown to influence human labor pain (Lee et al., 2020b), and deficiency of Kv9.3 disrupts parvalbumin interneuron physiology in mouse prefrontal cortex (Miyamae et al., 2021).”

(10) In general, the membrane potential used to activate Kv2 only channels and Kv2/KvS channels is too close to the activation V1/2. In case the comparing curves are displaced in their relative voltage dependence and voltage sensitivity, using that range of membrane potential may introduce a crucial error in the estimation of the conductance's relative amplitudes.

We now note that the relative conductances of Kv2-only vs Kv2/KvS channels are expected to vary with voltage protocol, as KvS inclusion results in channels with altered voltage responses.

Manuscript revisions:

“…the activation membrane potential is close to the half-maximal point of Kv2/KvS conductances. Thus the ratio of Kv2-like to KvS-like conductance is expected to vary with voltage protocols.”

(11) The use of tail currents to estimate conductance is problematic if (i) lack of current inactivation is not assured, and (ii) if the different currents, with possible different deactivation kinetics at the used membrane potential (e.g., mV), are not assured. Why was the activation peak used at times, and at different elapsed times the tail currents were used instead? These aspects of conductance's amplitude estimation methods should be well defined.

In CHO cells peak currents were analyzed because outward currents seem to offer the best signal/noise. In neurons, we restricted analysis to tail currents at elapsed times to minimize complications from non-Kv2 endogenous voltage-gated channels which deactivate more quickly. We have clarified this analysis in the methods section.

Manuscript revisions:

“In CHO cells peak currents were analyzed because outward currents seem to offer the best signal/noise. In neurons, we restricted analysis to tail currents at elapsed times to minimize complications from non-Kv2 endogenous voltage-gated channels which deactivate more quickly. In neurons, voltage gated currents remained in the toxin cocktail + RY785 and GxTX, that were sometimes unstable. To minimize complications from these currents, we restricted analysis of RY785 and GxTX subtraction experiments to tail currents at elapsed times to minimize complications from non-Kv2 endogenous voltage-gated channels which deactivate more quickly. We note that the analysis of conductance activation by using tail currents is only accurate when dealing with non-inactivating conductances. We expect that inactivation of Kv2/KvS conductances during the 200 ms pre-pulse is minimal (Salinas et al., 1997a; Kramer et al., 1998; Kerschensteiner and Stocker, 1999) and did not notice inactivation during the activation pulse. Also, deactivation kinetics can vary in a heterogenous population of Kv2/KvS heteromers. While analysis of tail currents could skew the quantification of total Kv2 like and KvS-like conductances, our data supports that mouse nociceptors and human neurons have tail currents that are resistant to RY785 and sensitive to GxTX consistent with the presence of Kv2/KvS heteromers.”

(12) Were the experiments including different conditions such as control, RY, and RY+GxTx done pair-wised? This could potentially better the statistics and strengthen the data and the conclusions drawn from them.

The control, RY, and RY+GxTX in neurons were done pairwise and the statistical tests performed for these experiments were pairwise tests. We have clarified this in the figure legends.

Manuscript revisions:

“Wilcoxon rank tests were paired, except the comparison of RY785 to vehicle which was unpaired.”

(13) The holding potential of the experiments, mostly -89 mV, may be biasing the estimation of Kv2 only channels vs. Kv2/KvS channels conductances. Figure 4I exemplifies this concern.

We agree. Figure 4I reveals that a holding potential of -89 mV vs -129 mV reduces conductance of Kv2.1/Kv8.1 heteromers vs Kv2.1 homomers in CHO cells by ~20%. We have now alerted readers that the ratio of Kv2 only channels vs. Kv2/KvS conductances can vary with holding voltage.

Manuscript revisions:

“Under these conditions, 58 ± 3 % (mean ± SEM) of the delayed rectifier conductance was resistant to RY785 yet sensitive to GxTX (KvS-like) (Fig 7 F). We note that the ratio of KvS- to Kv2-like conductances is expected to vary with holding potential, as KvS subunits can change the degree and voltage-dependence of steady state inactivation (e.g. Fig 4I).”

(14) It is possible that Figure 6A (control trace) and Figure 6C ("Kv2-like" trace) are the same, by mistake, since their noise pattern looks too similar.

Indeed the noise pattern of the Figure 6A (control trace) and Figure 6C ("Kv2-like" trace) are related, as they have inputs from the same trace, with Figure 6C ("Kv2-like" trace) being a subtraction of Figure 6A (+RY trace) from Figure 6A (control trace).

(15) For example, in Figure 7A, what is the identity of the current remaining after the RY+GxTx application? In Figure 7B, a supposed outlier in the group of data referring to "veh" in the right panel is what possibly is making this group different from +RY in the left panel (p=0.02, Wilcoxon rank test). I would recommend parametric tests only since the data is essentially quantitative.

In Figure 7A, we do not know the identity of the current remaining after the RY+GxTX application, the kinetics of the residual current appeared distinct from the Kv2/KvS-like currents blocked by RY or GxTX, but we did not analyze these.

The date in Figure 7B, was indeed the positive outlier in the group of data referring to "veh" in the right panel and contributes to the p-value, but we saw no reason to exclude it. We have now replaced the representative trace in 7B with a non-outlier trace. We respectfully disagree with the suggestion to use parametric statistical tests as we do not know the distribution underlying the variance our data.

Manuscript revisions:

“Subsequent application of 100 nM GxTX decreased tail currents by 68 ± 5% (mean ± SEM) of their original amplitude before RY785. We do not know the identity of the outward current that remains in the cocktail of inhibitors + RY785 + GxTX.”

(16) Please state the importance of using nonpeptidergic neurons to study silent Kv5.1 and Kv9.1 subunits. RNA data may not necessarily work to probe function or protein abundance, which is crucial in heteromeric complexes.

We have now more thoroughly explained our rationale for choosing the nonpeptidergic neurons.

RNA is not predictive of protein abundance, and we have not yet been successful in measuring KvS protein abundance in these neurons, so we've probed KvS abundance by assessing RY785 resistance.

Manuscript revisions:

“Mouse dorsal root ganglion (DRG) somatosensory neurons express Kv2 proteins (Stewart et al., 2024), have GxTX-sensitive conductances (Zheng et al., 2019), and express a variety of KvS transcripts (Bocksteins et al., 2009; Zheng et al., 2019), yet transcript abundance does not necessarily correlate with functional protein abundance. To record from a consistent subpopulation of mouse somatosensory neurons which has been shown to contain GxTXsensitive currents and have abundant expression of KvS mRNA transcripts (Zheng et al., 2019), we used a *MrgprdGFP* transgenic mouse line which expresses GFP in nonpeptidergic nociceptors (Zylka et al., 2005; Zheng et al., 2019). Deep sequencing identified that mRNA transcripts for Kv5.1, Kv6.2, Kv6.3, and Kv9.1 are present in GFP+ neurons of this mouse line (Zheng et al., 2019) and we confirmed the presence of Kv5.1 and Kv9.1 transcripts in GFP+ neurons from *MrgprdGFP* mice using RNAscope (Fig 7 Supplement 1).”

(17) In Figure 8B, were +RY data different from veh data? The figure shows no Wilcoxon (nonparametric) comparison and this is important to be stated. What conductance(s) is the vehicle solution blocking or promoting? What is RY dissolved in, DMSO? What is the DMSO final concentration?

We now state that in Figure 8B, +RY amplitudes were not statistically different from veh data in this limited data set. However, the RY-subtraction currents always had Kv2-like biophysical properties, whereas vehicle-subtraction currents had variable properties precluding biophysical analysis for Fig 8D.

In Figure 8B, we do not know what conductance(s) the vehicle solution is affecting, we think the changes observed are likely merely time dependent or due to the solution exchange itself. RY stock is in DMSO. All recording solutions have 0.1% DMSO final concentration, this is now noted in methods.

Manuscript revisions:

“Unlike mouse neurons, we did not detect a significant difference in tail currents of RY785 versus vehicle controls. However, RY785-subtracted currents always had Kv2-like biophysical properties whereas vehicle-subtraction currents had variable properties that precluded the same biophysical analysis. Overall, these results show that human DRG neurons can produce endogenous voltage-gated currents with pharmacology and gating consistent with Kv2/KvS heteromeric channels.”

“All RY785 solutions contained 0.1% DMSO. Vehicle control solutions also contained 0.1% DMSO but lacked RY785.”

(18) METHODS. The electrophysiology approach should be unified in all aspects as applicable and possible.

We have unified the mouse dorsal root ganglion and mouse superior cervical ganglion methods sections. We have kept CHO cells and mouse/human neurons section separate because the methods were substantially different.

(19) DISCUSSION. The discussion section spends half of its space trying to elaborate on possible blocking/inhibiting/modulating mechanisms for RY785. The present manuscript shows no data, at least not that I have noticed, that would evoke such discussion.

We have shortened this section, and enhance the discussion with structural models (new Fig 9), and our functional data indicating perturbed RY785 interaction with Kv2.1/8.1.

Manuscript revisions:

“In this pose, RY785 contacts a collection of Kv2.1 residues that vary in every KvS subtype (Fig 9 B,D,E). Notably, RY785 bound similarly to a 3:1 model of Kv2.1/Kv8.1, in contact with the three Kv2.1 subunits, yet avoided the Kv8.1 subunit (Fig 9C). This is consistent with RY785 binding less well to Kv2.1/Kv8.1 heteromers, and also suggests that a 3:1 Kv2:KvS channel could retain a RY785 binding site when open. However, the RY785 resistance of Kv2/KvS heteromers may primarily arise from perturbed interactions with the constricted central cavity of closed channels. In homomeric Kv2.1, RY785 becomes trapped in closed channels and prevents their voltage sensors from fully activating, indicating that RY785 must interact differently with closed channels (Marquis and Sack, 2022). Here we found that Kv2.1/Kv8.1 current rapidly recovers following washout of RY785, suggesting that Kv2.1/Kv8.1 heteromers do not readily trap RY785 (Figure 2 Supplement). Overall, the structural modeling suggests that KvS subunits sterically interfere with RY785 binding to the central cavity, while functional data suggest KvS subunits disrupt RY785 trapping in closed states.”

(20) DISCUSSION. Topics like ER retention and release upon certain conditions would be a better enrichment for the manuscript in my opinion.

ER retention of KvS subunits is indeed an important topic! However, we have opted not to delve into it here.

(21) DISCUSSION. Speculation about the binding site for RY on Kv2/KvS channels is also not touched by the data shown in the manuscript.

We have shortened this section of discussion, and now present this with structural models of RY785 docked to a Kv2.1 homomer and 3:1 Kv2.1: Kv8.1 heteromer (new Fig 9) to ground speculations. See manuscript changes noted in response to comment (19) above.

(22) DISCUSSION. An important reference is missing in regard to stoichiometry: Bocksteins et al., 2017. This work is the only one using a non-optical technique to add knowledge to that question.

Good point, and an excellent study we didn’t realize we’d not included before. We now include Bocksteins et al. 2017 as a reference in the Introduction.

(23) In my opinion, allosterism and orthosterism are concepts not yet useful for the discussion of RY binding sites without even a general piece of data.

We now include structural models of RY785 docked to a Kv2.1 homomer and 3:1 Kv2.1: Kv8.1 heteromer (new Fig 9) to ground blocking speculations. See manuscript changes noted in response to comment (19).

(24) The term "homogeneously susceptible" associated with a Hill slope close to 1 needs to be more elaborated.

Thank you, we have elaborated.

Manuscript revisions:

“Also, the degree of resistance to RY785 may vary if Kv2:KvS subunit stoichiometry varies. With high doses of RY785, we found that the concentration-response characteristics of Kv2.1/Kv8.1 in CHO cells revealed hallmarks of a homogenous channel population with a Hill slope close to 1 (Fig 2B). However, other KvS subunits might assemble in multiple stoichiometries and result in pharmacologically-distinct heteromer populations.”

(25) Stating the KvS are resistant to RY785 is not proper in my opinion. This opinion relates to the fact that the RY binding site in the channels is certainly not restricted to a binding site residing only on the Kv subunit.

Good point. We have now changed phrasing to convey that KvS subunits are a component of a heteromer that imbues RY785 resistance.

Manuscript revisions:

“These results show that voltage-gated outward currents in cells transfected with members from each KvS subtype have decreased sensitivity to RY785 but remain sensitive to GxTX. While we did not test every KvS subunit, the ubiquitous resistance suggests that all KvS subunits may provide resistance to 1 μM RY785 yet remain sensitive to GxTX, and that RY785 resistance is a hallmark of KvS-containing channels.”